# MVDiffusion: Enabling Holistic Multi-view Image Generation with Correspondence-Aware Diffusion

**Shitao Tang**[1*]    **Fuyang Zhang**[1*]    **Jiacheng Chen**[1]    **Peng Wang**[2]    **Yasutaka Furukawa**[1]

[1]Simon Fraser University    [2]Bytedance

*"This kitchen is a charming blend of rustic and modern, featuring a large reclaimed wood island with marble countertop, a sink surrounded by cabinets. A stainless-steel refrigerator stands tall. To the right of the sink, built-in wooden cabinets painted in a muted."*

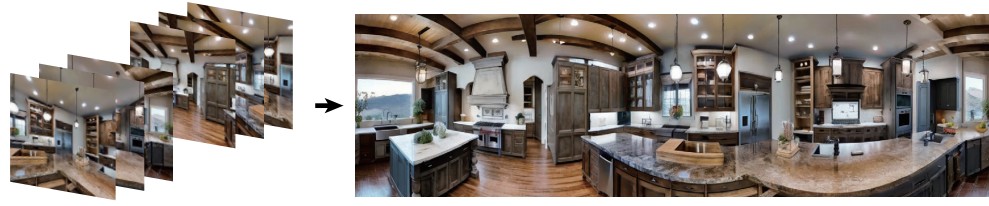

Consistent Multi-view Images          Closed-Loop Panoramic Image

*"A living room with multiple couches and a coffee table. A wooden book shelf filled with lots of books next to a door. A white refrigerator sitting next to a wooden bench."*

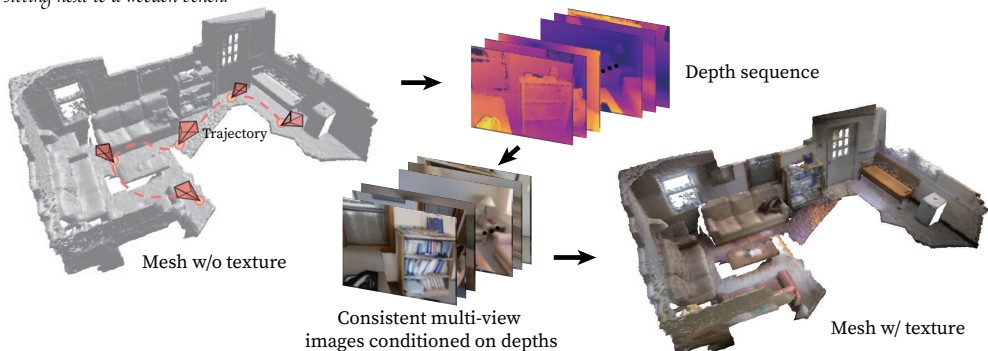

Figure 1: MVDiffusion synthesizes consistent multi-view images. *Top:* generating perspective crops which can be stitched into panorama; *Bottom:* generating coherent multi-view images from depths.

## Abstract

This paper introduces *MVDiffusion*, a simple yet effective method for generating consistent multi-view images from text prompts given pixel-to-pixel correspondences (*e.g.*, perspective crops from a panorama or multi-view images given depth maps and poses). Unlike prior methods that rely on iterative image warping and inpainting, MVDiffusion simultaneously generates all images with a global awareness, effectively addressing the prevalent error accumulation issue. At its core, MVDiffusion processes perspective images in parallel with a pre-trained text-to-image diffusion model, while integrating novel correspondence-aware attention layers to facilitate cross-view interactions. For panorama generation, while only trained with 10k panoramas, MVDiffusion is able to generate high-resolution photorealistic images for arbitrary texts or extrapolate one perspective image to a 360-degree view. For multi-view depth-to-image generation, MVDiffusion demonstrates state-of-the-art performance for texturing a scene mesh. The project page is at https://mvdiffusion.github.io/.

---

*Equal contribution. Contact the authors at shitaot@sfu.ca.

37th Conference on Neural Information Processing Systems (NeurIPS 2023).

# 1 Introduction

Photorealistic image synthesis aims to generate highly realistic images, enabling broad applications in virtual reality, augmented reality, video games, and filmmaking. The field has seen significant advancements in recent years, driven by the rapid development of deep learning techniques such as diffusion-based generative models [2, 16, 21, 39, 43, 44, 45].

One particularly successful domain is text-to-image generation. Effective approaches include generative adversarial networks [3, 12, 19], autoregressive transformers [10, 35, 49], and more recently, diffusion models [15, 17, 34, 37]. DALL-E 2 [34], Imagen [37] and others generate photorealistic images with large-scale diffusion models. Latent diffusion models [36] apply the diffusion process in the latent space, allowing more efficient computations and faster image synthesis.

Despite impressive progress, multi-view text-to-image synthesis still confronts issues of computational efficiency and consistency across views. A common approach involves an autoregressive generation process [5, 11, 18], where the generation of the $n$-th image is conditioned on the $(n-1)$-th image through image warping and inpainting techniques. However, this autoregressive approach results in accumulated errors and does not handle loop closure [11]. Moreover, the reliance on the previous image may pose challenges for complex scenarios or large viewpoint variations.

Our approach, dubbed MVDiffusion, generates multi-view images simultaneously, using multiple branches of a standard text-to-image model pre-trained on perspective images. Concretely, we use a stable diffusion (SD) model [36] and add a "correspondence-aware attention" (CAA) mechanism between the UNet blocks, which facilitates cross-view interactions and learns to enforce multi-view consistency. When training the CAA blocks, we freeze all the original SD weights to preserve the generalization capability of the pre-trained model.

In summary, the paper presents MVDiffusion, a multi-view text-to-image generation architecture that requires minimal changes to a standard pretrained text-to-image diffusion model, achieving state-of-the-art performance on two multi-view image generation tasks. For generating panorama, MVDiffusion synthesizes high-resolution photorealistic panoramic images given arbitrary per-view texts, or extrapolates one perspective image to a full 360-degree view. Impressively, despite being trained solely on a realistic indoor panorama dataset, MVDiffusion possesses the capability to create diverse panoramas, *e.g.* outdoor or cartoon style. For multi-view image generation conditioned on depths/poses, MVDiffusion demonstrates state-of-the-art performance for texturing a scene mesh.

# 2 Related Work

**Diffusion models.** Diffusion models [16, 21, 39, 42, 43, 44, 45] (DM) or score-based generative models are the essential theoretical framework of the exploding generative AI. Early works achieve superior sample quality and density estimation [8, 45] but require a long sampling trajectory. Advanced sampling techniques [20, 26, 40] accelerate the process while maintaining generation quality. Latent diffusion models [36] (LDMs) apply DM in a compressed latent space to reduce the computational cost and memory footprint, making the synthesis of high-resolution images feasible on consumer devices. We enable holistic multi-view image generation by the latent diffusion model.

**Image generation.** Diffusion Models (DM) dominate content generation. Foundational work such as DALL-E 2 [34], GLIDE [30], LDMs [36], and Imagen [37] have showcased significant capabilities in text-conditioned image generation. They train on extensive datasets and leverage the power of pre-trained language models. These large text-to-image Diffusion Models also establish strong foundations for fine-tuning towards domain-specific content generation. For instance, MultiDiffusion [1] and DiffCollage [53] failitates 360-degree image generation. However, the resulting images are not true panoramas since they do not incorporate camera projection models. Text2Light [6] synthesizes HDR panorama images from text using a multi-stage auto-regressive generative model. However, the leftmost and rightmost contents are not connected (i.e., loop closing).

**3D content generation.** Content generation technology has profound implications in VR/AR and entertainment industries, driving research to extend cutting-edge generation techniques from a single image to multiple images. Dreamfusion [31] and Magic3D [24] distill pre-trained Diffusion Models into a NeRF model [28] to generate 3D objects guided by text prompts. However, these works focus on objects rather than scenes. In the quest for scene generation, another approach [18] generates

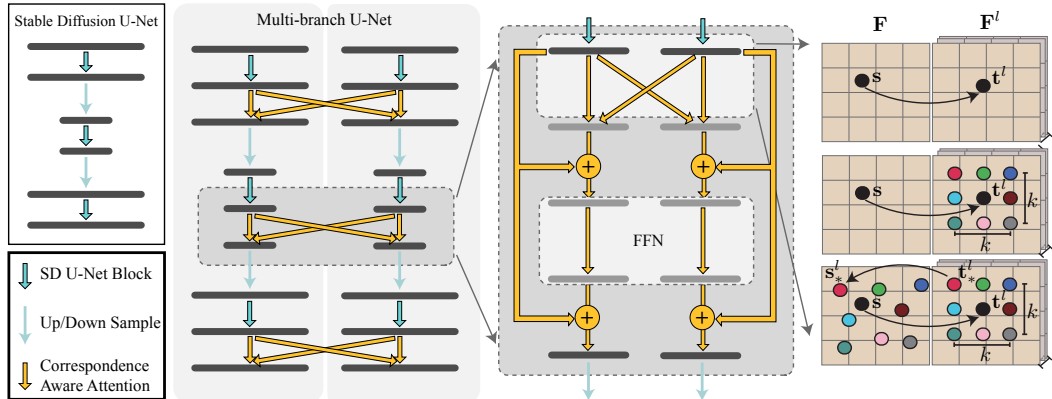

Figure 2: **MVDiffusion** generates multi-view images in parallel through the weight-sharing multi-branch UNet. To enforce multi-view consistency, the Correspondence-Aware Attention (CAA) block is inserted after each UNet block. "FFN" is an acronym for "Feed-Forward Network". The rightmost figure elaborates on the mechanisms of CAA.

prompt-conditioned images of indoor spaces by iteratively querying a pre-trained text-to-image Diffusion Model. SceneScape [11] generates novel views on zoom-out trajectories by employing image warping and inpainting techniques using diffusion models. Text2Room [18] adopts similar methods to generate a complete textured 3D room geometry. However, the generation of the $n$-th image is solely conditioned on the local context, resulting in accumulation errors and less favorable results. Our research takes a holistic approach and generates consistent multi-view images given camera poses and text prompts while fine-tuning pre-trained perspective-image Diffusion Models.

## 3  Preliminary

Latent Diffusion Models (LDM) [36] is the foundation of our method. LDM consists of three components: a variational autoencoder (VAE) [22] with encoder $\mathcal{E}$ and decoder $\mathcal{D}$, a denoising network $\epsilon_\theta$, and a condition encoder $\tau_\theta$.

High-resolution images $\mathbf{x} \in \mathbb{R}^{H \times W \times 3}$ are mapped to a low-dimensional latent space by $\mathbf{Z} = \mathcal{E}(\mathbf{x})$, where $\mathbf{Z} \in \mathbb{R}^{h \times w \times c}$. The down-sampling factor $f = H/h = W/w$ is set to 8 in the popular Stable Diffusion (SD). The latents are converted back to the image space by $\tilde{\mathbf{x}} = \mathcal{D}(\mathbf{Z})$.

The LDM training objective is given as

$$L_{LDM} := \mathbb{E}_{\mathcal{E}(\mathbf{x}), \mathbf{y}, \boldsymbol{\epsilon} \sim \mathcal{N}(0,1), t} \left[ \| \boldsymbol{\epsilon} - \epsilon_\theta(\mathbf{Z}_t, t, \tau_\theta(\mathbf{y})) \|_2^2 \right], \tag{1}$$

where $t$ is uniformly sampled from 1 to $T$ and $\mathbf{Z}_t$ is the noisy latent at time step $t$. The denoising network $\epsilon_\theta$ is a time-conditional UNet [8], augmented with cross-attention mechanisms to incorporate the optional condition encoding $\tau_\theta(\mathbf{y})$. $\mathbf{y}$ could be a text-prompt, an image, or any other user-specified condition.

At sampling time, the denoising (reverse) process generates samples in the latent space, and the decoder produces high-resolution images with a single forward pass. Advanced samplers [20, 26, 40] can further accelerate the sampling process.

## 4  MVDiffusion: Holistic Multi-view Image Generation

MVDiffusion generates multiple images simultaneously by running multiple copies/branches of a stable diffusion model with a novel inter-branch "correspondence-aware attention" (CAA) mechanism to facilitate multi-view consistency. Figure 2 presents an overview of multi-branch UNet and the CAA designs. The system is applicable when pixel-to-pixel correspondences are available between images, specifically for cases of 1) Generating a panorama or extrapolating a perspective image to a panorama. The panorama consists of perspective images sharing the camera center where

pixel-to-pixel correspondences are obtained by planar tomography and 2) Texture mapping of a given geometry where multiple images of arbitrary camera poses establish pixel-to-pixel correspondences through depth-based unprojection and projection. We first introduce panorama generation (§4.1), which employs generation modules, and then multi-view depth-to-image generation (§4.2), which employs generation and interpolation modules. Since the interpolation module does not contain CAA blocks, §4.1 will also cover the design of the CAA block and explain how it is inserted into the multi-branch UNet.

## 4.1 Panorama generation

In MVDiffusion, a panorama is realized by generating eight perspective views, each possessing a horizontal field of view of $90°$ with a $45°$ overlap. To achieve this, we generate eight $512 \times 512$ images by the generation module using a frozen pretrained stable diffusion model [47].

**Generation module.** The proposed module generates eight $512 \times 512$ images. It accomplishes this through a process of simultaneous denoising. This process involves feeding each noisy latent into a shared UNet architecture, dubbed as the multi-branch UNet, to predict noises concurrently. In order to ensure multi-view consistency, a correspondence-aware attention (CAA) block is introduced following each UNet block. The CAA block follows the final ResNet blocks and is responsible for taking in multi-view features and fusing them together.

**Correspondence-aware attention (CAA).** The CAA block operates on $N$ feature maps concurrently, as shown in Figure 2. For the i-th source feature map, denoted as $\mathbf{F}$, it performs cross-attention with the remaining $(N-1)$ target feature maps, represented as $\mathbf{F}^l$.

For a token located at position $(\mathbf{s})$ in the source feature map, we compute a message based on the corresponding pixels $\{\mathbf{t}^l\}$ in the target feature maps $\{\mathbf{F}^l\}$ (not necessarily at integer coordinates) with local neighborhoods. Concretely, for each target pixel $\mathbf{t}^l$, we consider a $K \times K$ neighborhood $\mathcal{N}(\mathbf{t}^l)$ by adding integer displacements $(d_x/d_y)$ to the (x/y) coordinate, where $|d_x| < K/2$ and $|d_y| < K/2$. In practice, we use $K = 3$ with a neighborhood of 9 points.

$$\mathbf{M} = \sum_l \sum_{t_*^l \in \mathcal{N}(\mathbf{t}^l)} \text{SoftMax}\left(\left[\mathbf{W_Q}\bar{\mathbf{F}}(\mathbf{s})\right] \cdot \left[\mathbf{W_K}\bar{\mathbf{F}}^l(t_*^l)\right]\right)\mathbf{W_V}\bar{\mathbf{F}}^l(t_*^l), \tag{2}$$

$$\bar{\mathbf{F}}(\mathbf{s}) = \mathbf{F}(\mathbf{s}) + \boldsymbol{\gamma}(0), \quad \bar{\mathbf{F}}^l(t_*^l) = \mathbf{F}^l(t_*^l) + \boldsymbol{\gamma}(\mathbf{s}_*^l - \mathbf{s}). \tag{3}$$

The message $\mathbf{M}$ calculation follows the standard attention mechanism that aggregates information from the target feature pixels $\{t_*^l\}$ to the source $(s)$. $\mathbf{W_Q}$, $\mathbf{W_K}$, and $\mathbf{W_V}$ are the query, key and value matrices. The key difference is the added position encoding $\gamma(\cdot)$ to the target feature $\mathbf{F}^l(t_*^l)$ based on the 2D displacement between its corresponding location $\mathbf{s}_*^l$ in the source image and $\mathbf{s}$. The displacement provides the relative location in the local neighborhood. Note that a displacement is a 2D vector, and we apply a standard frequency encoding [50] to the displacement in both x and y coordinates, then concatenate. A target feature $\mathbf{F}^l(t_*^l)$ is not at an integer location and is obtained by bilinear interpolation. To retain the inherent capabilities of the stable diffusion model [47], we initialize the final linear layer of the transformer block and the final convolutional layer of the residual block to be zero, as suggested in ControlNet [52]. This initialization strategy ensures that our modifications do not disrupt the original functionality of the stable diffusion model.

**Panorama extraporlation.** The goal is to generate full 360-degree panoramic views (seven target images) based on a single perspective image (one condition image) and the per-view text prompts. We use SD's impainting model [48] as the base model as it takes one condition image. Similar to the generation model, CAA blocks with zero initializations are inserted into the UNet and trained on our datasets.

For the generation process, the model reinitializes the latents of both the target and condition images with noises from standard Gaussians. In the UNet branch of the condition image, we concatenate a mask of ones to the image (4 channels in total). This concatenated image then serves as the input to the inpainting model, which ensures that the content of the condition image remains the same. On the contrary, in the UNet branch for a target image, we concatenate a black image (pixel values of zero) with a mask of zeros to serve as the input, thus requiring the inpainting model to generate a completely new image based on the text condition and the correspondences with the condition image.

**Training.** We insert CAA block into the pretrained stable diffusion Unet [47] or stable diffusion impainting Unet [48] to ensure multi-view consistency. The pretrained network is frozen while we use the following loss to train the CAA block:

$$L_{\text{MVDiffusion}} := \mathbb{E}_{\left\{\mathbf{Z}_t^i = \mathcal{E}(\mathbf{x}^i)\right\}_{i=1}^N, \{\boldsymbol{\epsilon}^i \sim \mathcal{N}(0,I)\}_{i=1}^N, \mathbf{y}, t} \left[ \sum_{i=1}^N \|\boldsymbol{\epsilon}^i - \epsilon_\theta^i(\left\{\mathbf{Z}_t^i\right\}, t, \tau_\theta(\mathbf{y}))\|_2^2 \right]. \quad (4)$$

### 4.2 Multiview depth-to-image generation

The multiview depth-to-image task aims to generate multi-view images given depths/poses. Such images establish pixel-to-pixel correspondences through depth-based unprojection and projection. MVDiffusion's process starts with the generation module producing key images, which are then densified by the interpolation module for a more detailed representation.

**Generation module.** The generation module for multi-view depth-to-image generation is similar to the one for panorama generation. The module generates a set of $192 \times 256$ images. We use depth-conditioned stable diffusion model [46] as the base generation module and simultaneously generate multi-view images through a multi-branch UNet. The CAA blocks are adopted to ensure multi-view consistency.

**Interpolation module.** The interpolation module of MVDiffusion, inspired by VideoLDM [2], creates $N$ images between a pair of 'key frames', which have been previously generated by the generation module. This model utilizes the same UNet structure and correspondence attention weights as the generation model, with extra convolutional layers, and it reinitializes the latent of both the in-between images and key images using Gaussian noise. A distinct feature of this module is that the UNet branch of key images is conditioned on images already generated. Specifically, this condition is incorporated into every UNet block. In the UNet branch of key images, the generated images are concatenated with a mask of ones (4 channels), and then a zero convolution operation is used to downsample the image to the corresponding feature map size. These downsampled conditions are subsequently added to the input of the UNet blocks. For the branch of in-between images, we take a different approach. We append a black image, with pixel values of zero, to a mask of zeros, and apply the same zero convolution operation to downsample the image to match the corresponding feature map size. These downsampled conditions are also added to the input of the UNet blocks. This procedure essentially trains the module such that when the mask is one, the branch regenerates the conditioned images, and when the mask is zero, the branch generates the in-between images.

**Training.** we adopt a two-stage training process. In the first stage, we fine-tune the SD UNet model using all ScanNet data. This stage is single-view training (Eq. 1) without the CAA blocks. In the second stage, we integrate the CAA blocks, and the image condition blocks into the UNet, and only these added parameters are trained. We use the same loss as panorama generation to train the model.

## 5 Experiments

We evaluate MVDiffusion on two tasks: panoramic image generation and multi-view depth-to-image generation. We first describe implementation details and the evaluation metrics.

**Implementation details.** We have implemented the system with PyTorch while using publicly available Stable Diffusion codes from Diffusers [51]. The model consists of a denoising UNet to execute the denoising process within a compressed latent space and a VAE to connect the image and latent spaces. The pre-trained VAE of the Stable Diffusion is maintained with official weights and is used to encode images during the training phase and decode the latent codes into images during the inference phase. We have used a machine with 4 NVIDIA RTX A6000 GPUs for training and inference. Specific details and results of these varying configurations are provided in the corresponding sections.

**Evaluation matrics.** The evaluation metrics cover two aspects, image quality of generated images and their consistency.

• *Image quality* is measured by Fréchet Inception Distance (FID) [14], Inception Score (IS) [38], and CLIP Score (CS) [32]. FID measures the distribution similarity between the features of the generated

Prompt: "a living room with a large glass table, white chairs and a giant window. A TV is attached to the wall and a fancy crystal chandelier is hanging on the ceiling."

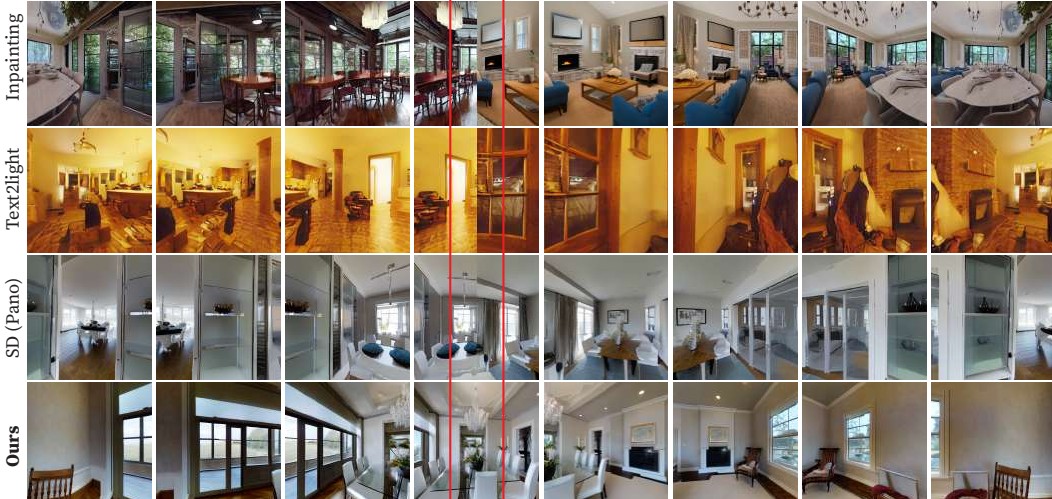

Figure 3: Qualitative evaluation for panorama generation. The red box indicates the area stitched leftmost and rightmost content. More results are available in the supplementary material.

and real images. The Inception Score is based on the diversity and predictability of generated images. CLIP Score measures the text-image similarity using pretrained CLIP models [33].

• *Multi-view consistency* is measured by the metric based on pixel-level similarity. The area of multi-view image generation is still in an early stage, and there is no common metric for multi-view consistency. We propose a new metric based on Peak Signal-to-Noise Ratio (PSNR). Concretely, given multi-view images, we compute the PSNR between all the overlapping regions and then compare this "overlapping PSNR" for ground truth images and generated images. The final score is defined as the ratio of the "overlapping PSNR" of generated images to that of ground truth images. Higher values indicate better consistency.

The rest of the section explains the experimental settings and results more, while the full details are referred to the supplementary.

## 5.1 Panoramic image generation

This task generates perspective crops covering the panoramic field of view, where the challenge is to ensure consistency in the overlapping regions. Matterport3D [4] is a comprehensive indoor scene dataset that consists of 90 buildings with 10,912 panoramic images. We allocate 9820 and 1092 panoramas for training and evaluation, respectively.

**Baselines.** We have selected three related state-of-the-art methods for thorough comparisons. The details of the baselines are briefly summarized as follows (full implementation details can be found in the appendix):

• *Text2Light*[6] creates HDR panoramic images from text using a multi-stage auto-regressive generative model. To obtain homographic images, we project the generated panoramas into perspective images using the same camera settings (FoV=90°, rotation=45°).
• *Stable Diffusion (SD)*[36] is a text-to-image model capable of generating high-quality perspective images from text. For comparison, we fine-tuned Stable Diffusion using panoramic images and then extracted perspective images in a similar manner.
• *Inpainting methods* [11, 18] operate through an iterative process, warping generated images to the current image and using an inpainting model to fill in the unknown area. Specifically, we employed the inpainting model from Stable Diffusion v2 [36] for this purpose.

**Results.** Table 1 and Figure 3 present the quantitative and qualitative evaluations, respectively. We then discuss the comparison between MVDiffusion and each baseline:

Prompt: "Majestically rising towards the heavens, the snow-capped mountain stood, its jagged peaks cloaked in a shroud of ethereal clouds, its rugged slopes a stark contrast against the serene azure sky, and its silent grandeur exuding an air of ancient wisdom and timeless solitude."

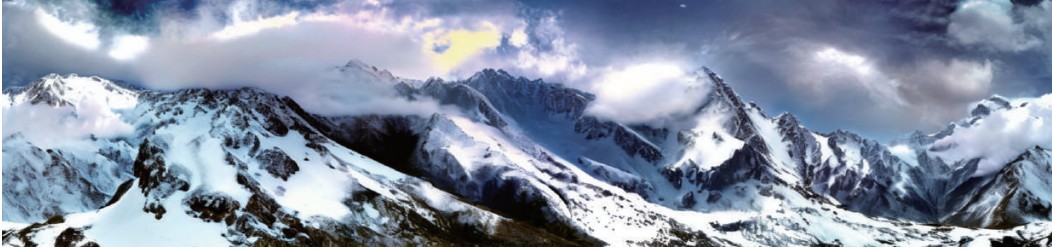

Figure 4: Example of panorama generation of outdoor scene. More results are available in the supplementary material.

Prompt: "A large kitchen with a center island and white canbinets. A dining room with tables and chairs. A room with a lot of windows and a wooden floor."

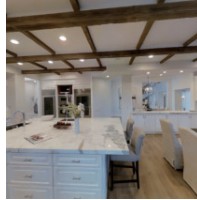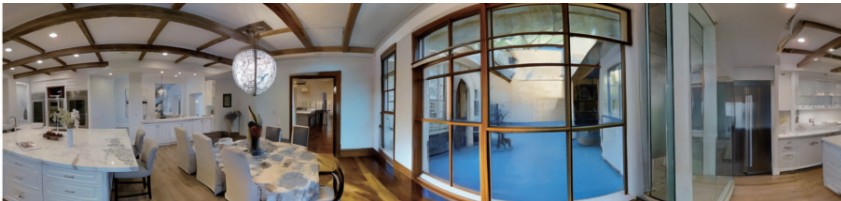

Figure 5: Image&text-conditioned panorama generation results. More results are available in the supplementary material.

• *Compared with Text2Light[6]*: Text2Light is based on auto-regressive transformers and shows low FID, primarily because diffusion models perform generally better. Another drawback is the inconsistency between the left and the right panorama borders, as illustrated in Figure 3.

• *Compared with Stable Diffusion (panorama)*[36]: MVDiffusion obtain higher IS, CS, and FID than SD (pano). Like Text2light, this model also encounters an issue with inconsistency at the left and right borders. Our approach addresses this problem by enforcing explicit consistency through correspondence-aware attention, resulting in seamless panoramas. Another shortcoming of this model is its requirement for substantial data to reach robust generalization. In contrast, our model, leveraging a frozen pre-trained stable diffusion, demonstrates a robust generalization ability with a small amount of training data, as shown in Figure 4.

| Method | FID ↓ | IS ↑ | CS ↑ |
|---|---|---|---|
| Impainting [11, 18] | 42.13 | 7.08 | 29.05 |
| Text2light [6] | 48.71 | 5.41 | 25.98 |
| SD (Pano) [36] | 23.02 | 6.58 | 28.63 |
| SD (Perspective) [36] | 25.59 | 7.29 | 30.25 |
| MVDiffusion(Ours) | **21.44** | **7.32** | **30.04** |

Table 1: Quantitative evaluation with Fréchet Inception Distance (FID), Inception Score (IS), and CLIP Score (CS).

• *Compared with inpainting method* [11, 18]: Inpainting methods also exhibit worse performance due to the error accumulations, as evidenced by the gradual style change throughout the generated image sequence in Figure 3.

• *Compared with Stable Diffusion (perspective)*[36]: We also evaluate the original stable diffusion on perspective images of the same Matterport3D testing set. This method cannot generate multi-view images but is a good reference for performance comparison. The results suggest that our method does not incur performance drops when adapting SD for multi-view image generation.

**Generate images in the wild.** Despite our model being trained solely on indoor scene data, it demonstrates a remarkable capability to generalize across various domains. This broad applicability is maintained due to the original capabilities of stable diffusion, which are preserved by freezing the stable diffusion weights. As exemplified in Figure 4, we stich the perspective images into a

panorama and show that our MVDiffusion model can successfully generate outdoor panoramas. Further examples illustrating the model's ability to generate diverse scenes, including those it was not explicitly trained on, can be found in the supplementary materials.

**Image&text-conditioned panorama generation.** In Figure 5, we show one example of image&text-conditioned panorama generation. MVDiffsion demonstrates the extraordinary capability of extrapolating the whole scene based on one perspective image.

Prompt: "A kitchen with white cabinets and white appliances. A kitchen with a white stove top oven next to a sink. A cutting board on the counter."

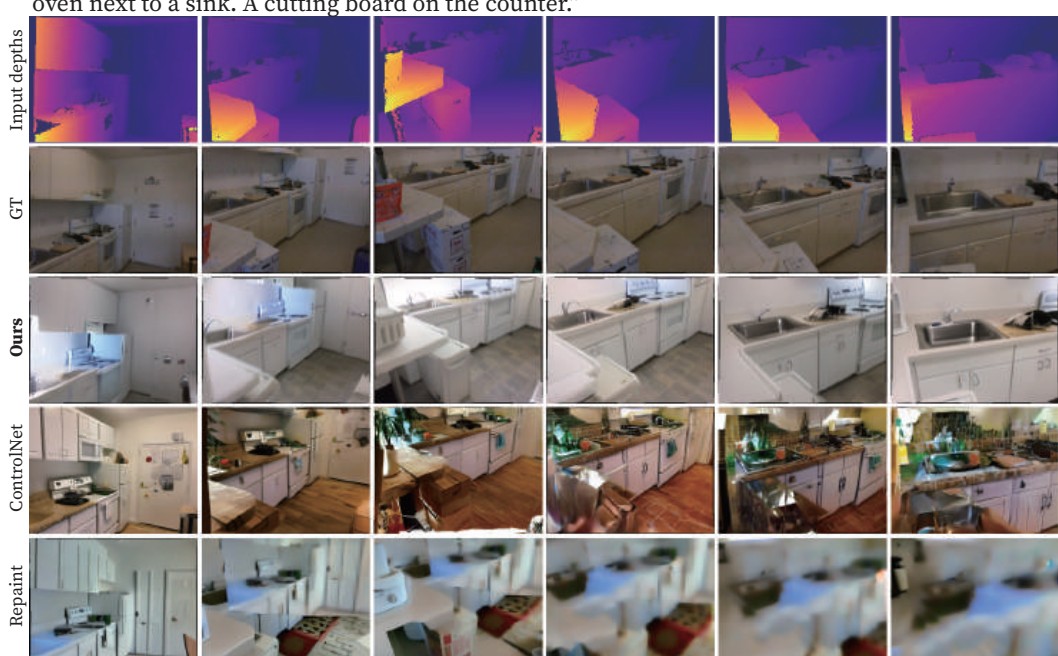

Figure 6: Qualitative evaluation for depth-to-image generation. More results are available in the supplementary material.

## 5.2 Multi view depth-to-image generation

This task converts a sequence of depth images into a sequence of RGB images while preserving the underlying geometry and maintaining multi-view consistency. ScanNet is an indoor RGB-D video dataset comprising over 1513 training scenes and 100 testing scenes, all with known camera parameters. We train our model on the training scenes and evaluate it on the testing

| Method | FID ↓ | IS ↑ | CS ↑ |
|---|---|---|---|
| RePaint [27] | 70.05 | 7.15 | 26.98 |
| ControlNet [52] | 43.67 | 7.23 | 28.14 |
| Ours | **23.10** | **7.27** | **29.03** |

Table 2: Comparison in Fréchet Inception Distance (FID), Inception Score (IS), and CLIP Score (CS) for multiview depth-to-image generation.

scenes. In order to construct our training and testing sequences, we initially select key frames, ensuring that each consecutive pair of key frames has an overlap of approximately 65%. Each training sample consists of 12 sequential keyframes. For evaluation purposes, we conduct two sets of experiments. For our quantitative evaluation, we have carefully chosen 590 non-overlapping image sequences from the testing set, each composed of 12 individual images. In terms of qualitative assessment, we first employ the generation model to produce all the key frames within a given test sequence. Following this, the image&text-conditioned generation model is utilized to enrich or densify these images. Notably, even though our model has been trained using a frame length of 12, it has the capability to be generalized to accommodate any number of frames. Ultimately, we fuse the RGBD sequence into a cohesive scene mesh.

**Baselines.** To our knowledge, no direct baselines exist for scene-level depth-to-image generation. Some generate 3D textures for object meshes, but often require complete object geometry to optimize the generated texture from many angles [5, 29]. This is unsuitable for our setting where geometry is provided for the parts visible in the input images. Therefore, we have selected two baselines:

| Task → | Panorama | | Depth-to-image | |
|---|---|---|---|---|
| Method | PSNR ↑ | Ratio ↑ | PSNR ↑ | Ratio ↑ |
| G.T. | 37.7 | 1.00 | 21.41 | 1.00 |
| SD (Perspective) | 10.6 | 0.28 | 11.20 | 0.44 |
| MVDiffusion (Ours) | **25.4** | **0.67** | **17.41** | **0.76** |

Table 3: Multi-view consistency for panorama generation and multi-view depth-to-image generation.

• *RePaint*[27]: This method uses an image diffusion model for inpainting holes in an image, where we employ the depth2image model from Stable Diffusion v2[36]. For each frame, we warp the generated images to the current frame and employ the RePaint technique [27] to fill in the holes. This baseline model is also fine-tuned on our training set.

• *Depth-conditioned ControlNet*: We train a depth-conditioned ControlNet model combined with the inpainting Stable Diffusion method with the same training set as ours. The implementation is based on a public codebase [7]. This model is capable of image inpainting conditioned on depths. We use the same pipeline as the above method.

**Results.** Table 2, Figure 6, Figure 7, and Figure 8 present the quantitative and qualitative results of our generation models. Our approach achieves a notably better FID, as shown in Table 2. As depicted in Figure 6, the repaint method generates progressively blurry images, while ControlNet produces nonsensical content when the motion is large. These issues arise since both methods depend on partial results and local context during inpainting, causing error propagation. Our method overcomes these challenges by enforcing global constraints with the correspondence-aware attention and generating images simultaneously. Figure 7 exhibits the keyframes at the left and the right, where the intermediate frames are generated in the middle, which are consistent throughout the sequence. Figure 8 illustrates the textured scene meshes produced by our method.

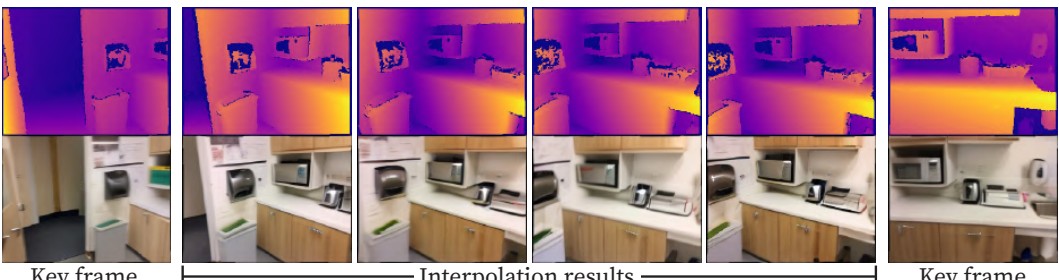

Key frame    |———————— Interpolation results ————————|    Key frame

Figure 7: Visualization of image&text-conditioned generated frames. We interpolate 4 frames (second image to fifth image). More visualization results are presented in supplementary materials.

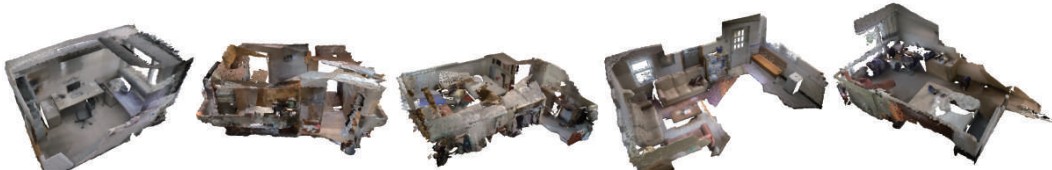

Figure 8: Mesh visualization. MVDiffusion first generates RGB images given depths/poses and then fuse them into mesh with TSDF fusion.

### 5.3 Measuring multi-view consistency

The multi-view consistency is evaluated with the metric as explained earlier. For panoramic image generation, we select image pairs with a rotation angle of 45 degrees and resize them to $1024 \times 1024$. For multi-view depth-to-image generation, consecutive image pairs are used and resized to $192 \times 256$. PSNR is computed among all pixels within overlapping regions for panoramic image generation. For

multi-view depth-to-image generation, a depth check discards pixels with depth errors above $0.5m$, the PSNR is then computed on the remaining overlapping pixels.

**Results.** In Table 3, we first use the real images to set up the upper limit, yielding a PSNR ratio of 1.0. We then evaluate our generation model without the correspondence attention (i.e., an original stable diffusion model), effectively acting as the lower limit. Our method, presented in the last row, achieves a PSNR ratio of 0.67 and 0.76 for the two tasks respectively, confirming an improved multi-view consistency.

## 6    Conclusion

This paper introduces MVDiffusion, an innovative approach that simultaneously generates consistent multi-view images. Our principal novelty is the integration of a correspondence-aware attention (CAA) mechanism, which ensures cross-view consistency by recognizing pixel-to-pixel correspondences. This mechanism is incorporated into each UNet block of stable diffusion. By using a frozen pretrained stable diffusion model, extensive experiments show that MVDiffusion achieves state-of-the-art performance in panoramic image generation and multi-view depth-to-image generation, effectively mitigating the issue of accumulation error of previous approaches. Furthermore, our high-level idea has the potential to be extended to other generative tasks like video prediction or 3D object generation, opening up new avenues for the content generation of more complex and large-scale scenes.

**Limitations.** The primary limitation of MVDiffusion lies in its computational time and resource requirements. Despite using advanced samplers, our models need at least 50 steps to generate high-quality images, which is a common bottleneck of all DM-based generation approaches. Additionally, the memory-intensive nature of MVDiffusion, resulting from the parallel denoising limits its scalability. This constraint poses challenges for its application in more complex applications that require a large number of images (*e.g.*, long virtual tour).

**Broader impact.** MVDiffusion enables the generation of detailed environments for video games, virtual reality experiences, and movie scenes directly from written scripts, vastly speeding up production and reducing costs. However, like all techniques for generating high-quality content, our method might be used to produce disinformation.

**Acknowledgements.** This research is partially supported by NSERC Discovery Grants with Accelerator Supplements and DND/NSERC Discovery Grant Supplement, NSERC Alliance Grants, and John R. Evans Leaders Fund (JELF). We thank the Digital Research Alliance of Canada and BC DRI Group for providing computational resources.

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

# Appendix:
## MVDiffusion: Enabling Holistic Multi-view Image Generation with Correspondence-Aware Diffusion

The appendix provides 1) the full architecture specification of correspondence attention; 2) the implementation details of the MVDiffusion system; and 3) additional experimental results in the same format as the figures in the main paper.

## A    Network Architecture of correspondence-aware attention block

The architecture of correspondence-aware attention block is similar to vision transformers [9], with the inclusion of zero convolutions as suggested in ControlNet [52] and GELU [13] activation function. $C$, $H$, $W$ are channel numbers, height and width respectively.

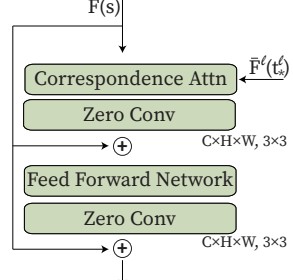

Figure 9: The architecture of the correspondence-aware attention block.

## B    Implementation details of MVDiffusion

### B.1    Panorama image generation

**Training and inference details.** The generation model in our approach is built upon Stable-diffusion-v2 [47]. We train the model on perspective images with a resolution of $512 \times 512$ for 10 epochs. The training is performed using the AdamW optimizer with a batch size of 4 and a learning rate of $2e^{-4}$, utilizing four A6000 GPUs. During inference, we utilize the DDIM sampler with a step size of 50 to perform parallel denoising of the eight generated images. Additionally, we employ blip2 [23] to generate texts for each perspective image, and during both training and inference, we use the corresponding prompts.

### B.2    Implementation details of baselines

We introduce implementation details of baseline in the following.

- *Text2Light* [6] We combine the prompts of each perspective image and use the released pretrained model to generate the panorama.
- *Stable Diffusion (panorama)*[36] We fine-tuned Stable Diffusion using the panorama images within our training dataset, which contains 9820 panorama images at resolusion $512 \times 1024$. We fine-tuned the UNet layer of the Stable diffusion while keeping VAE layers frozen. We use AdamW optimizer with a learning rate of $1e^{-6}$ and batch size is 4, utilizing four A6000 GPUs.
- *Inpainting methods* [11, 18] In our approach, we employ Stable-diffusion-v2 [47] to generate the first image in the sequence based on the corresponding text prompt. For each subsequent image in the sequence, we utilize image warping to align the previous image with the current image. The warped image is then passed through Stable-diffusion-inpaint [48] to fill in the missing regions and generate the final image.
- *Stable diffusion (perspective)* In our approach, we utilize the model trained in the first stage of the generation module to generate the perspective images. During testing, each perspective image is associated with its own text prompt.

### B.3    Geometry conditioned image generation

**Training and inference details.** Our generation model is derived from the stable-diffusion-2-depth framework [46]. In the initial phase, we fine-tune the model on all the perspective images of ScanNet dataset at a resolution of $192 \times 256$ for 10 epochs. This training process employs the AdamW optimizer [25] with a learning rate of $1e^{-5}$ and a batch size of 256, utilizing four A6000 GPUs. In the second stage, we introduce the correspondence-aware attention block and extra resizing convolutional layers for image conditioned generation. The training data consists of two categories: 1) generation training data, selected from 12 consecutive frames with overlaps of around 0.65, and 2) interpolation training data, derived from randomly chosen pairs of consecutive key frames and ten intermediate

frames. During each training epoch, we maintain a careful balance by randomly sampling data from both these categories in a 1:1 ratio. The correspondence-aware attention blocks and additional convolution layers are subsequently trained for 20 epochs, with a batch size of 4 and a learning rate of $1e^{-4}$, using the same four A6000 GPUs. During the inference stage, we employ the DDIM [41] sampler with a step size of 50 to perform parallel denoising on eight images.

## B.4 Implementation details of baselines

We introduce the implementation details of baselines in the following.

• *RePaint*[27]: In our method, we utilize depth-conditioned Stable-diffusion-v2 [47] to generate the first image in the sequence. For each subsequent image, we condition it on the previous image by applying latent warping. This helps align the generated image with the previous one. To complete the remaining areas of the image, we employ the Repaint technique [27] for inpainting.

• *Depth-conditioned ControlNet*: We use the same method to generate the first image as the above method. Next, we warp the generated images to the current frame and use Stable-inpainting model [48] to fill the hole. To incorporate depth information into the inpainting model, we utilize a method from a public codebase [7], which adds the feature from depth-conditioned ControlNet [52] into each UNet layer of the inpainting model. For more detailed information, please refer to their code repository. In order to reduce the domain gap, the Stable-inpainting model has been fine-tuned on our training dataset. Similar to other fine-tuning procedures, we only fine-tuned UNet layers while keeping VAE part fixed. The fine-tuning was conducted on a machine with four A6000 GPUs. The batch size is 4 and the learning rate is $1e^{-6}$. We used AdamW as the optimizer. During inference, we utilize the DDIM sampler with a step size of 50 for generating images.

## B.5 Visualization results

Figures 10-17 present supplementary results for panorama generation. In these figures, we showcase the output panorama images generated by both Stable diffusion (panorama) and Text2light methods. To compare the consistency between the left and right borders, we apply a rotation to the border regions, bringing them towards the center of the images. These additional visualizations provide further insights into the quality and alignment of the generated panorama images.

Figures 18-19 show additional results for image- & text-conditioned panorama generation. MVDiffusion extrapolates the whole scene based on one provided perspective image and text description.

Figure 20 provides additional results for generalization to out-of-training distribution data. Our model is only trained on MP3D indoor data while showing a robust generalization ability (e.g. outdoor scene).

Figures 21-26 show additional results with two baseline methods (depth-conditioned ControlNet [52] and Repaint [27]).

Figure 27 shows additional results of interpolated frames. The keyframes are at the left and the right, the middle frames are generated by applying our Interpolation module (see Sec. 4.2 in the main paper). The consistency is maintained throughout the whole sequence.

A bedroom with a large bed and sliding glass doors. An open door to a patio with an ocean view. A room with a chair and a lamp.

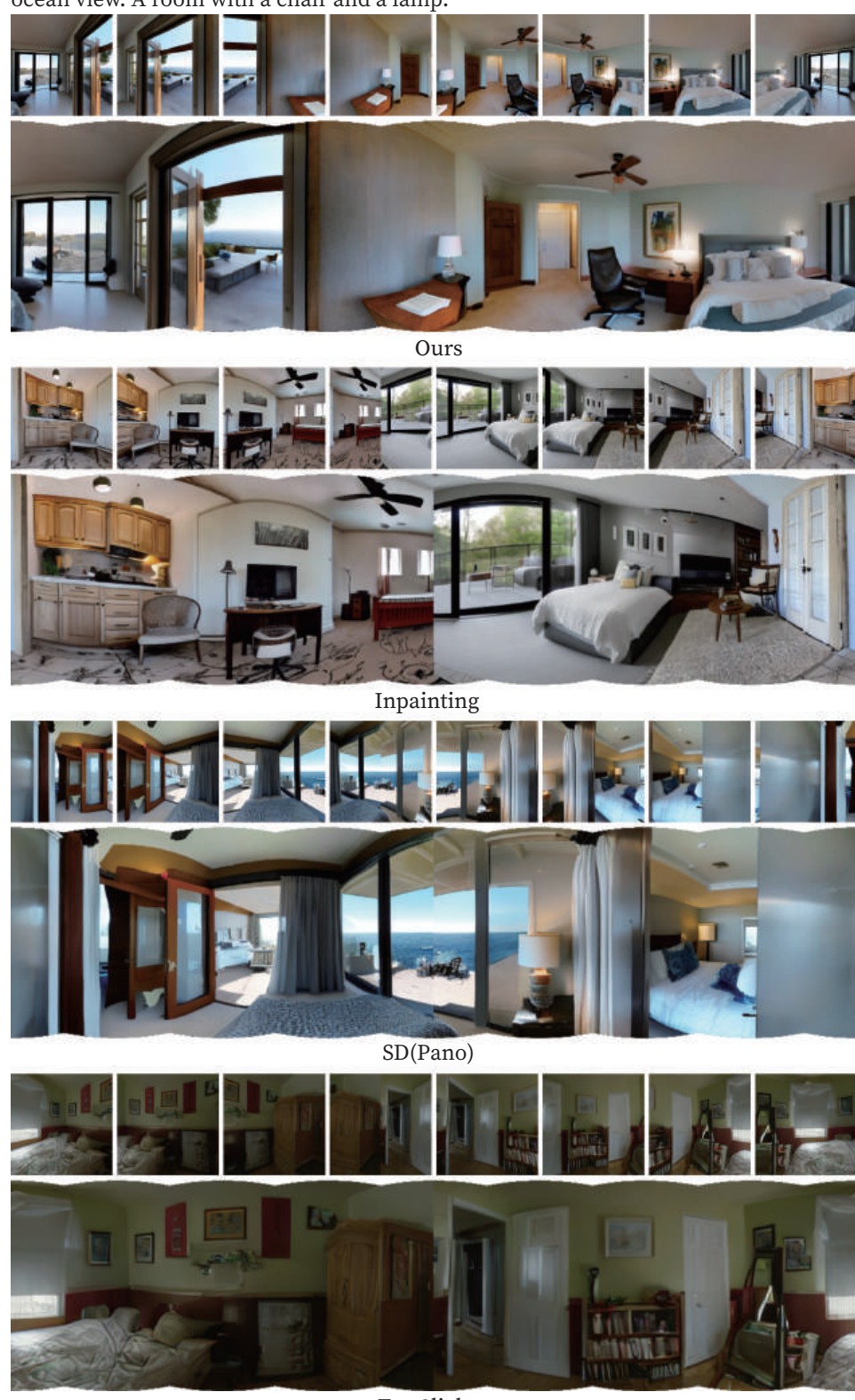

Ours

Inpainting

SD(Pano)

Text2light

Figure 10: Addition results for panorama generation

A room with white cabinets and a wooden floor. A walk-in closet with lots of shelves.
An empty room with a closet and shelves.

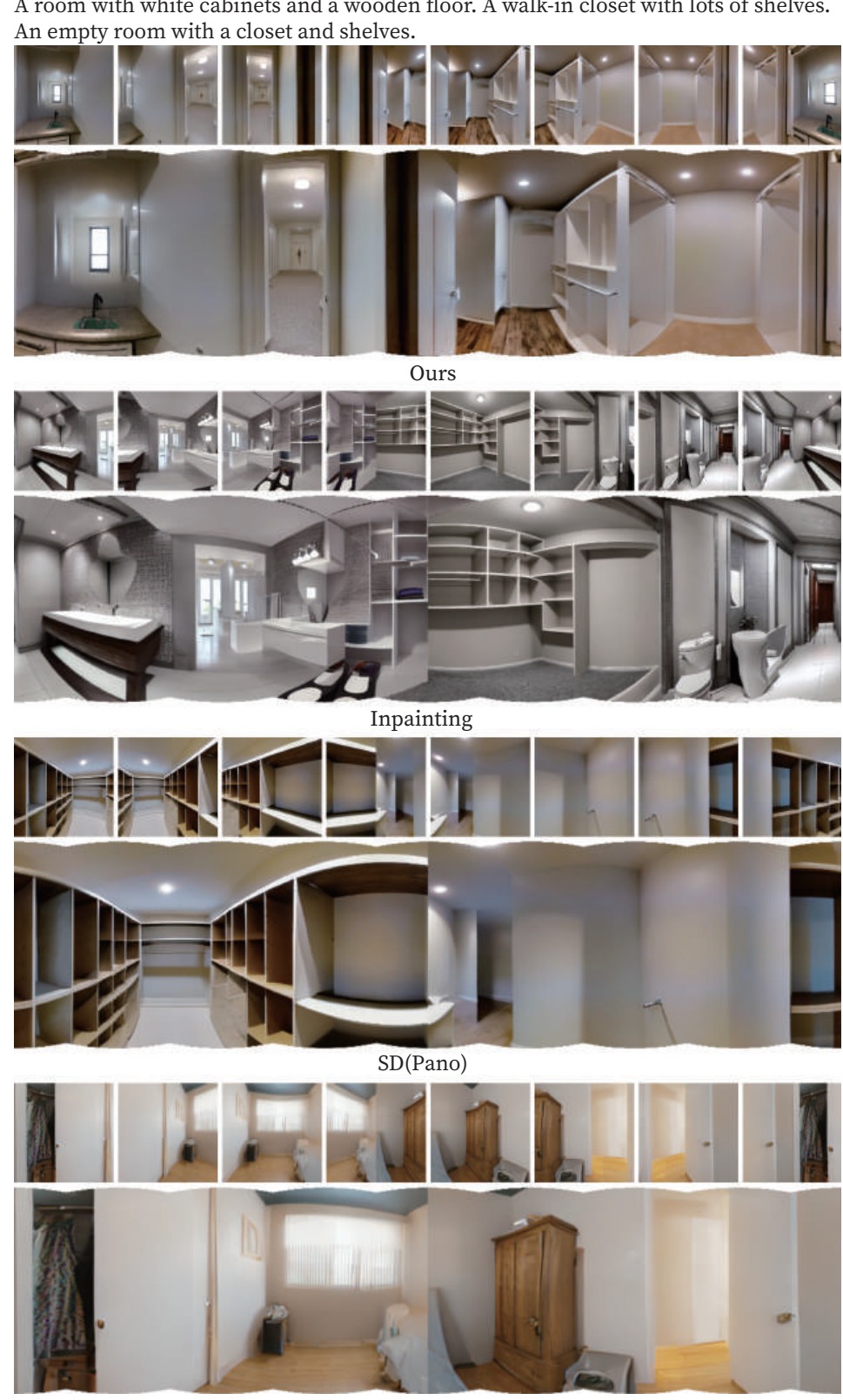

Ours

Inpainting

SD(Pano)

Text2light

Figure 11: Addition results for panorama generation

A living room filled with furniture, a grand piano, a fire place, a painting , a large window. A living room with a couch and a ceiling fan.

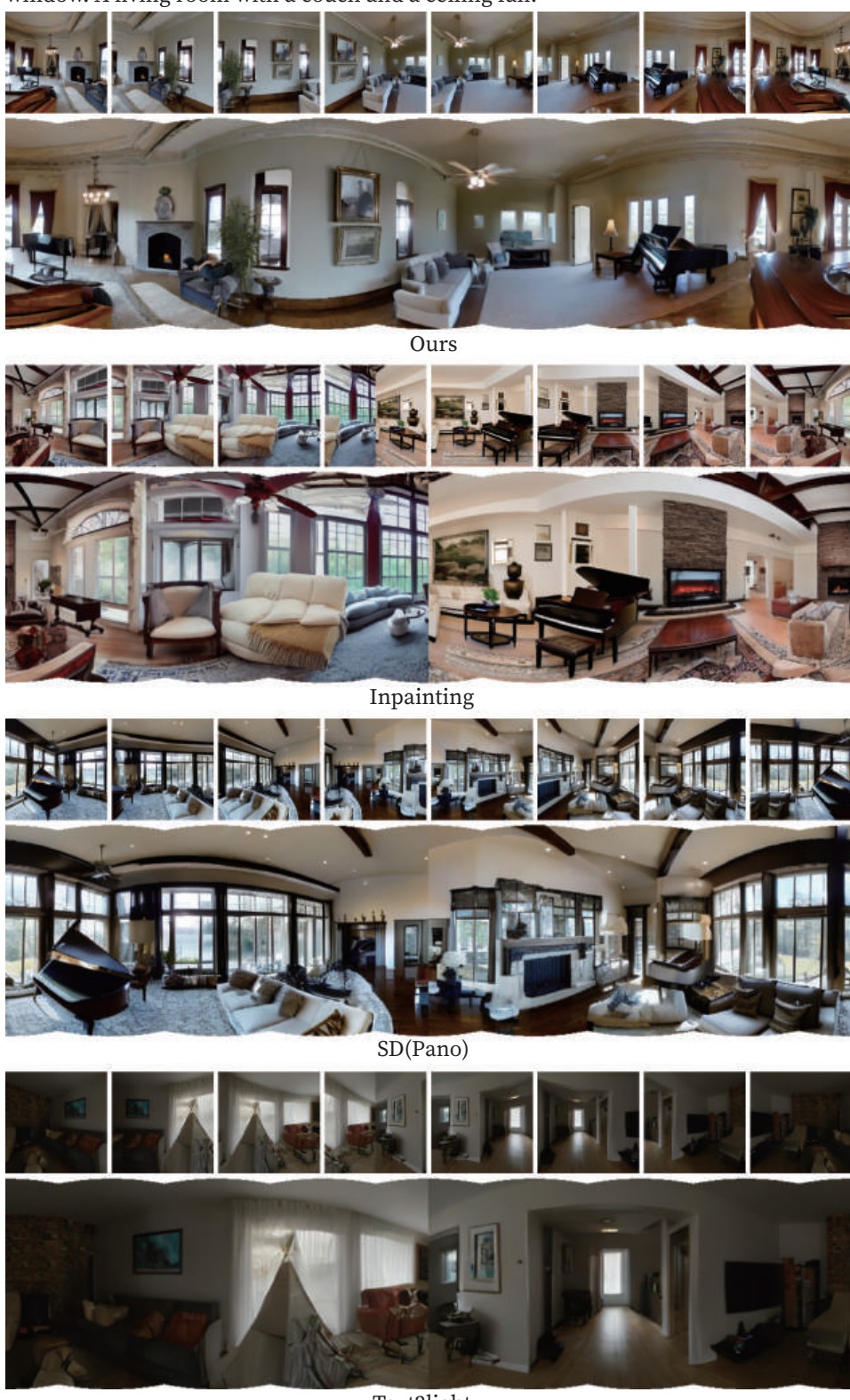

Ours

Inpainting

SD(Pano)

Text2light

Figure 12: Addition results for panorama generation

A grand piano sitting in a living room next to a window. A living room filled with furniture and a fire place. A black piano sitting in a living room next to a window.

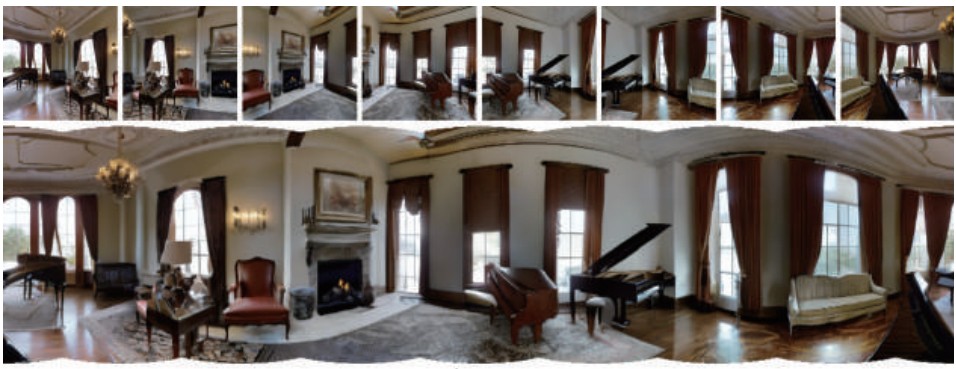

Ours

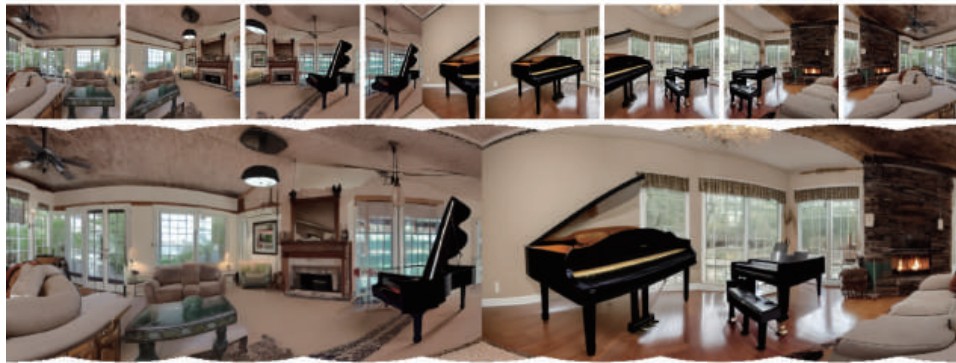

Inpainting

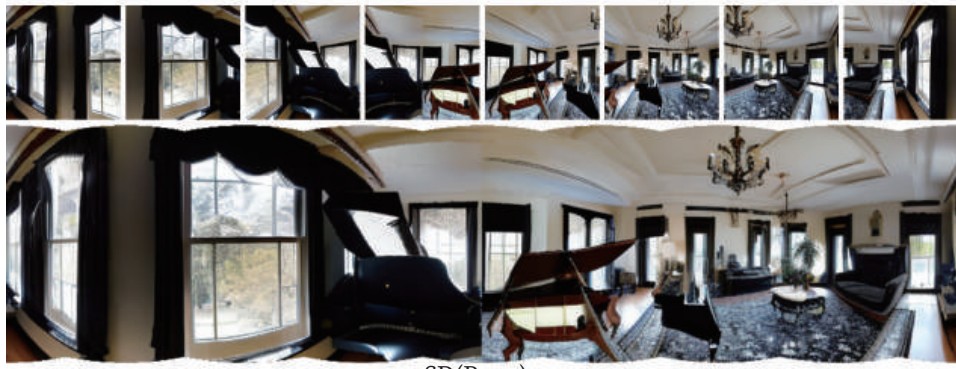

SD(Pano)

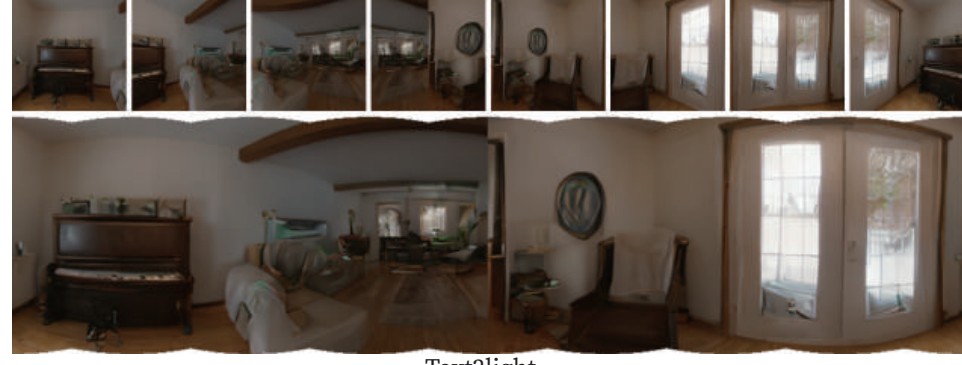

Text2light

Figure 13: Addition results for panorama generation

A dining room with a chandelier a table and chairs. A living room with white walls and wood floors. A room with a mirror on the wall.

Ours

Inpainting

SD(Pano)

Text2light

Figure 14: Addition results for panorama generation

A bedroom with a bed and a mirror and a window. A vase of flowers on a shelf in a room. A hallway with two framed pictures on the wall.

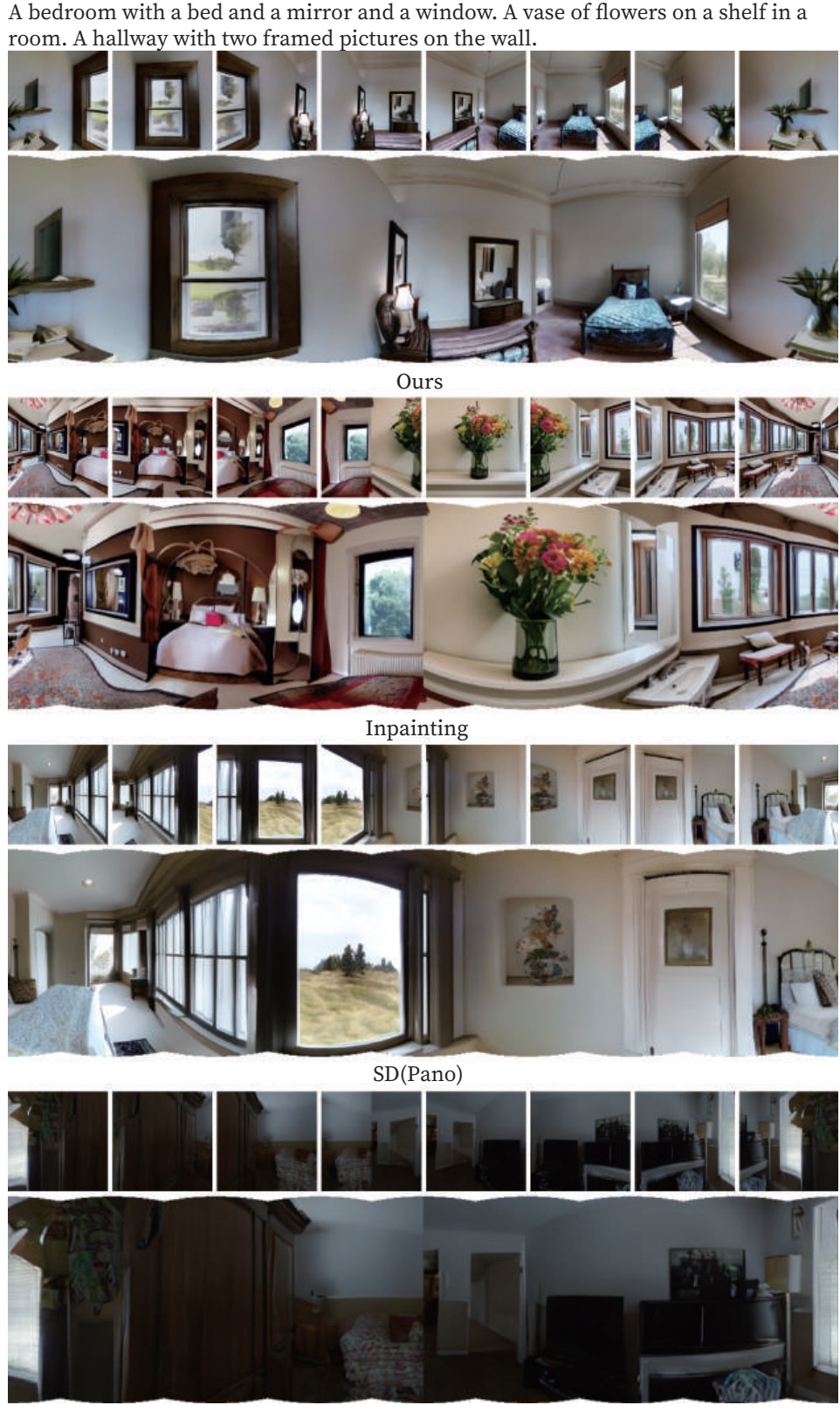

Ours

Inpainting

SD(Pano)

Text2light

Figure 15: Addition results for panorama generation

A living room filled with furniture and a large mirror. A table with a lamp on it. A living room filled with furniture and a chandelier.

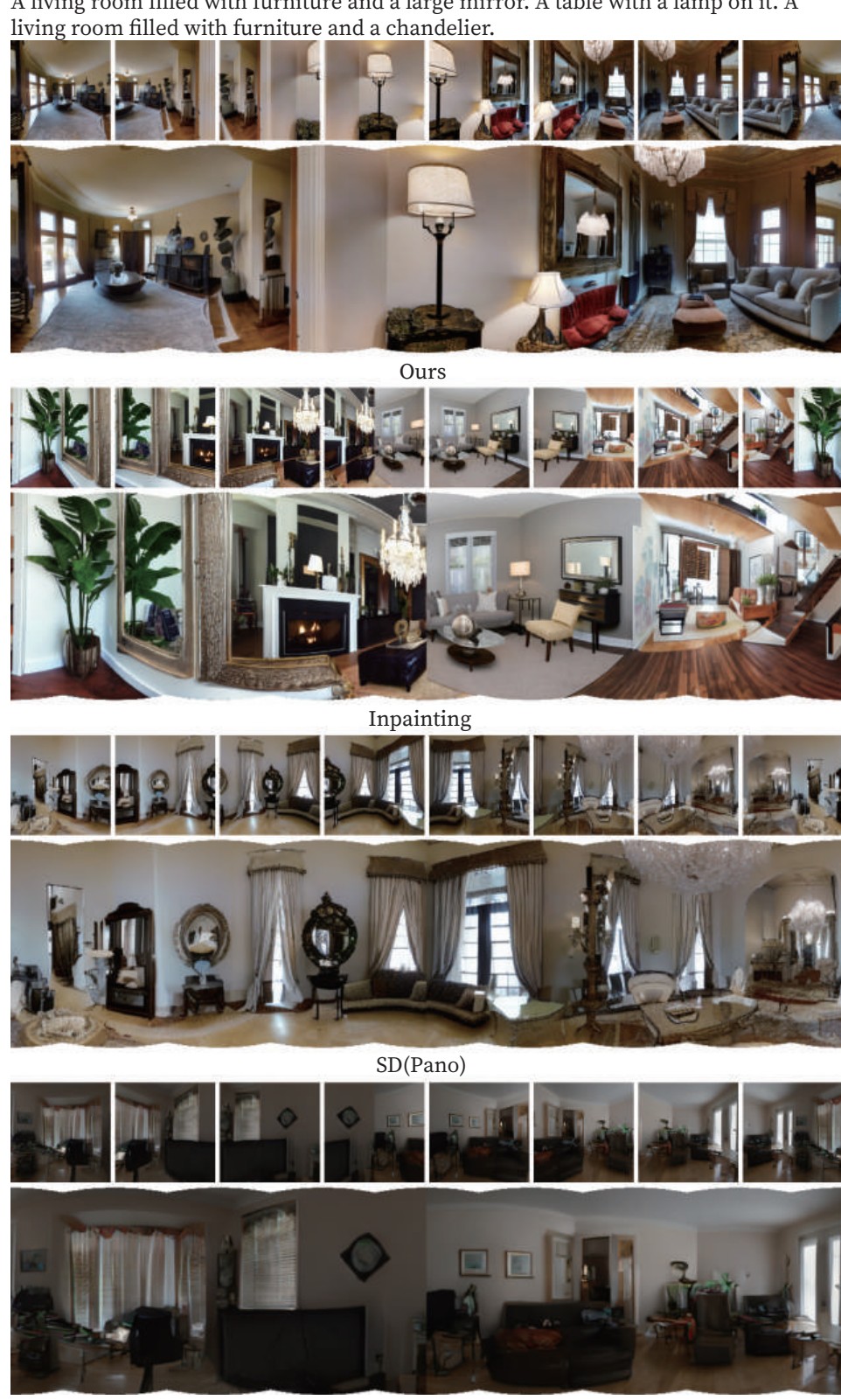

Ours

Inpainting

SD(Pano)

Text2light

Figure 16: Addition results for panorama generation

A large kitchen with a center island and white canbinets. A dining room with a table and chairs. A view of a pool through a glass door. A room with a lot of windows.

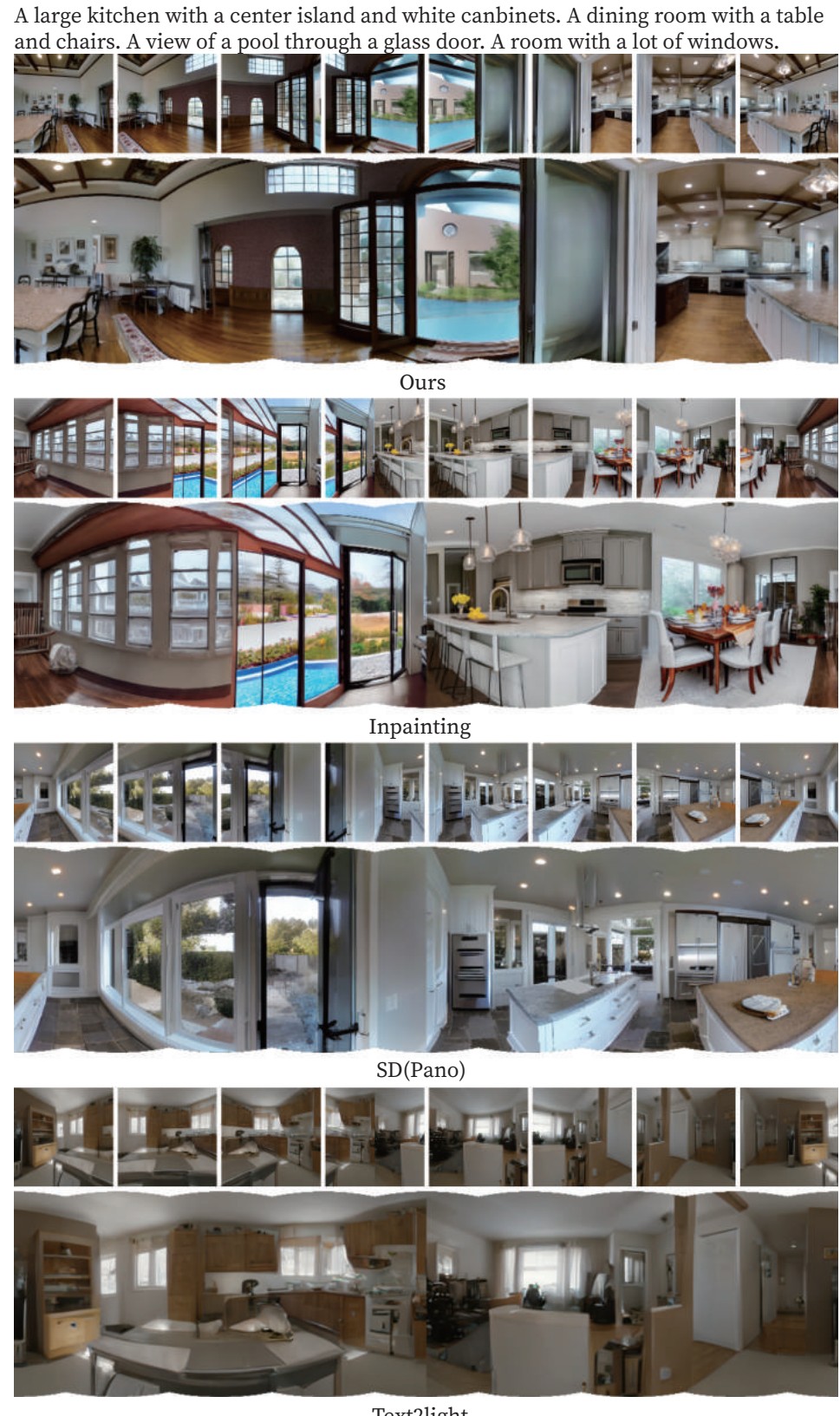

Ours

Inpainting

SD(Pano)

Text2light

Figure 17: Addition results for panorama generation

A living room with a white couvh and wooden floors. A living room filled with furniture, a ceiling fan and a flat screen TV.

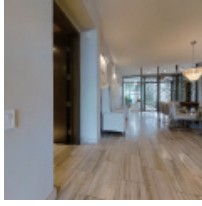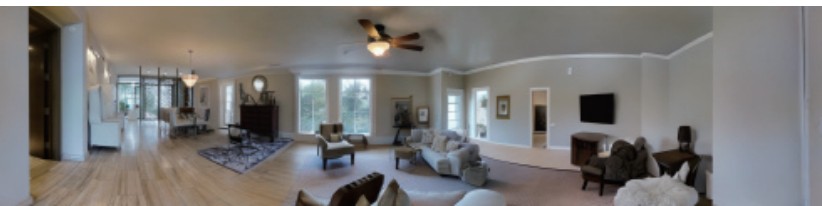

A bathroom with a bathtub and a window. A large bathroom with a wooden floor and two sinks and a large mirror.

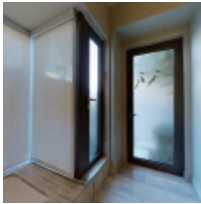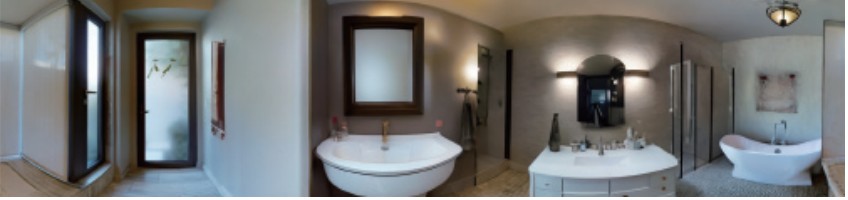

A potted plant in front of a window. A living room filled with furniture and a piano. A view of a patio through an open door.

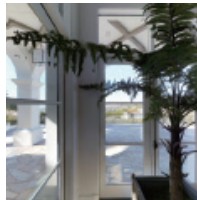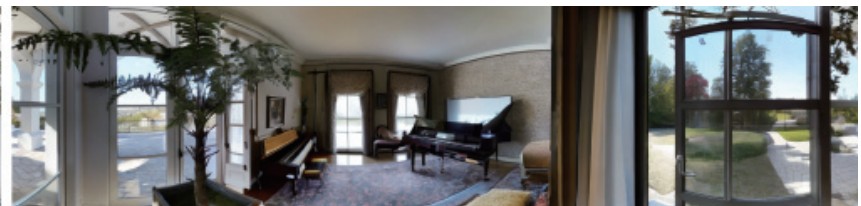

A hallway heading to a living room. A hallway with a chandelier and white walls. A large white cabinet with morrors on it.

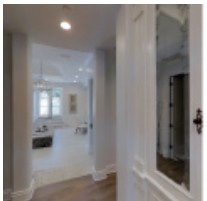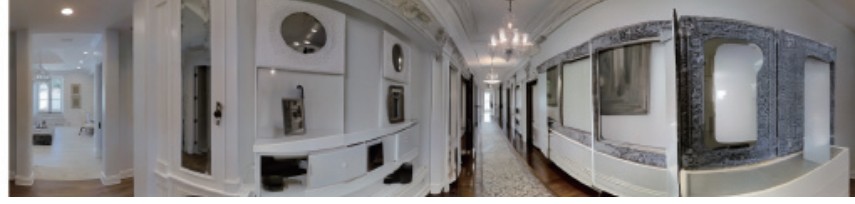

A room with a large window and a painting on the wall. A dining room with a table, chairs and a chandelier. A view of a kitchen with white canbinets and wood floors.

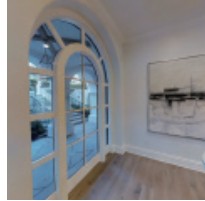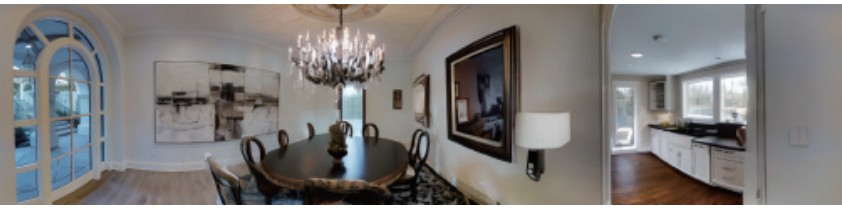

Figure 18: Addition results for dual-conditioned generation

A bedroom with a bed and a window. A bedroom with a couch and a bed. A vase of flowers sitting next to a window on top of a table.

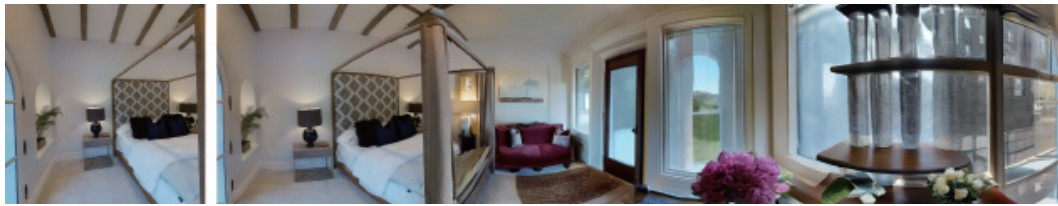

A bathroom with a mirror and a vase of flowers. A bathroom with a toilet and a shower stall.

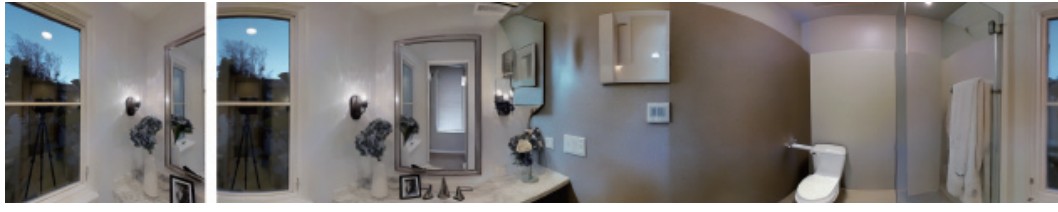

A room with a sink and a mirror. A couple of beds in a room with a clock on the wall. A hotel room with a bed and a desk.

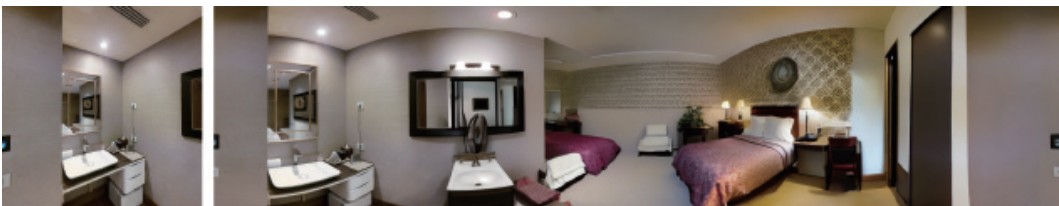

A theater room with a row of recliners. A hallway leading to a movie theater room. A large empty room with a projector screen in the corner.

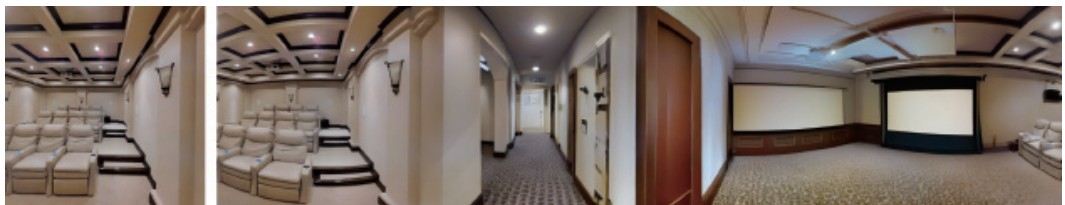

A house with a fire pit in front of it. A patio with a fire pit in the middle of it. A large house with a pool in front of it. A swimming pool with a waterfall and a rock wall.

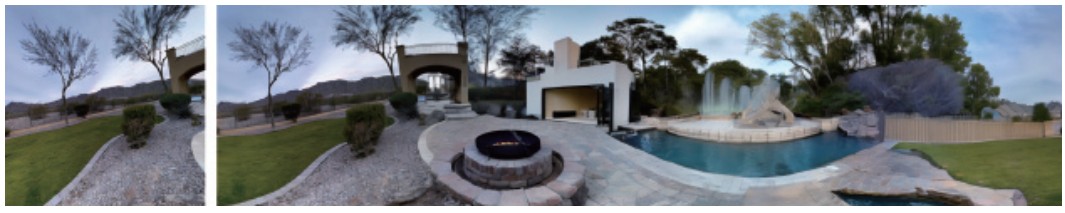

Figure 19: Addition results for dual-conditioned generation

Bursting with vibrant hues and exaggerated proportions, the cartoon-styled room sparkled with whimsy and cheer, with floating shelves crammed with oddly shaped trinkets, a comically oversized polka-dot armchair perched near a gravity-defying, tilted lamp, and the candy-striped wallpaper creating a playful backdrop to the merry chaos, exuding a sense of fun and boundless imagination.

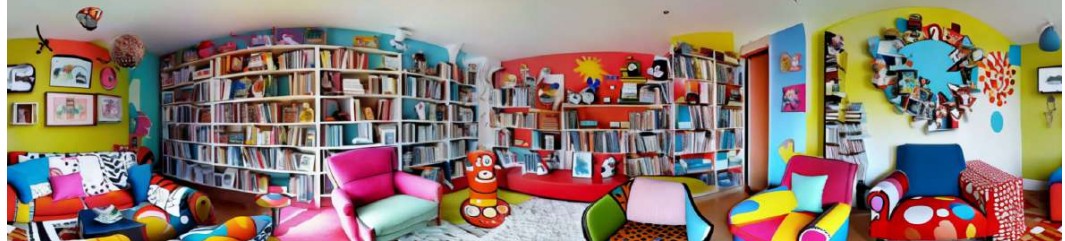

Bathed in the soft, dappled light of the setting sun, the silent street lay undisturbed, revealing the grandeur of its cobblestone texture, the rusted lampposts bearing witness to forgotten stories, and the ancient, ivy-clad houses standing stoically, their shuttered windows and weather-beaten doors speaking volumes about their passage through time.

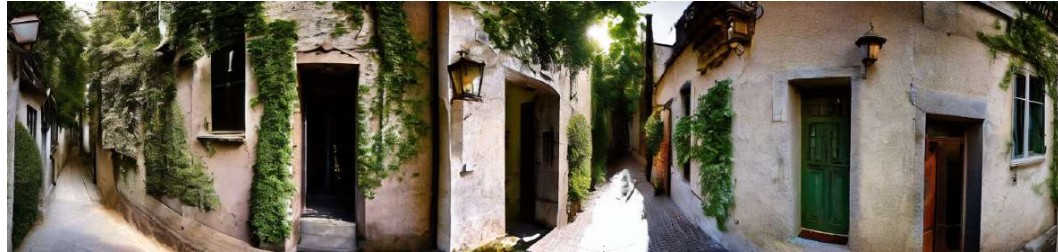

Awash with the soothing hues of an array of blossoms, the tranquil garden was a symphony of life and color, where the soft murmur of the babbling brook intertwined with the whispering willows, and the iridescent petals danced in the gentle breeze, creating an enchanting sanctuary of beauty and serenity.

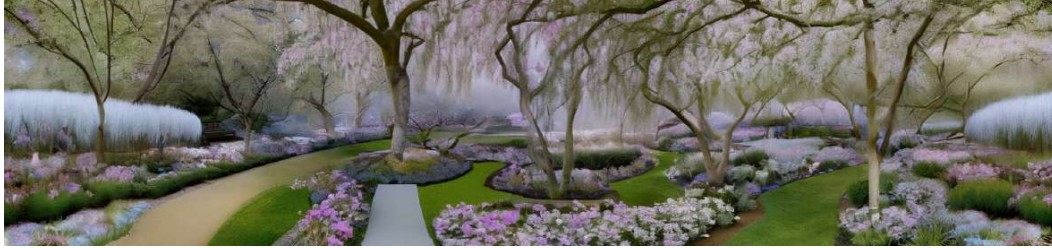

This ancient castle stands majestically on a rugged hill, with towering turrets, thick stone walls, and a formidable drawbridge over a time-worn moat. Inside, dimly lit hallways lead to grand chambers adorned with ornate carvings, all resonating with the echoes of a bygone era.

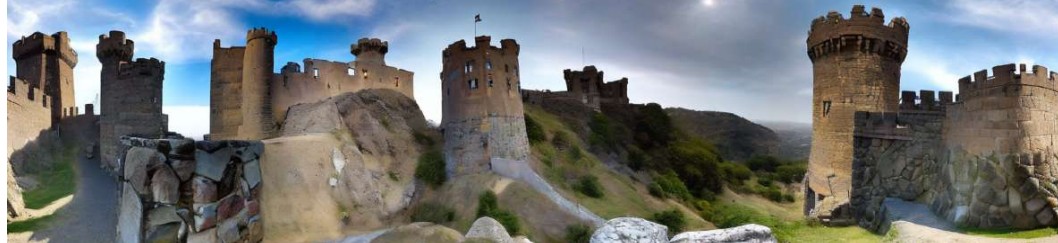

Figure 20: Additional results for generalization to out-of-training distribution data.

A white refrigerator freezer sitting next to a window. A trash can next to a refrigerator in a kitchen.

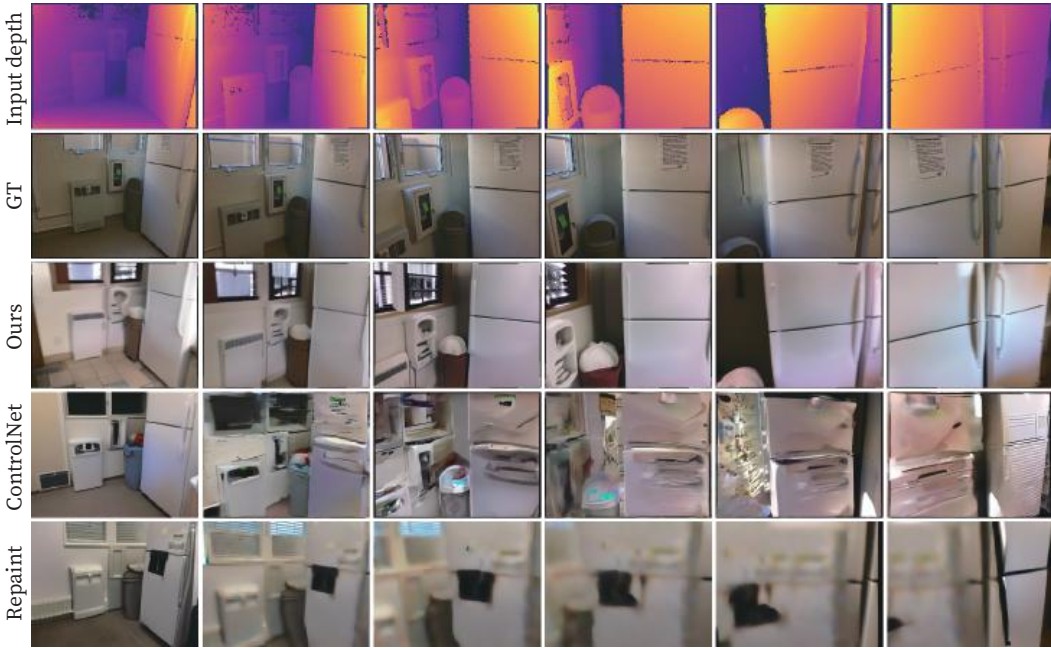

A living room filled with furniture and a piano. A living room with a couch, chair and pictures on the wall.

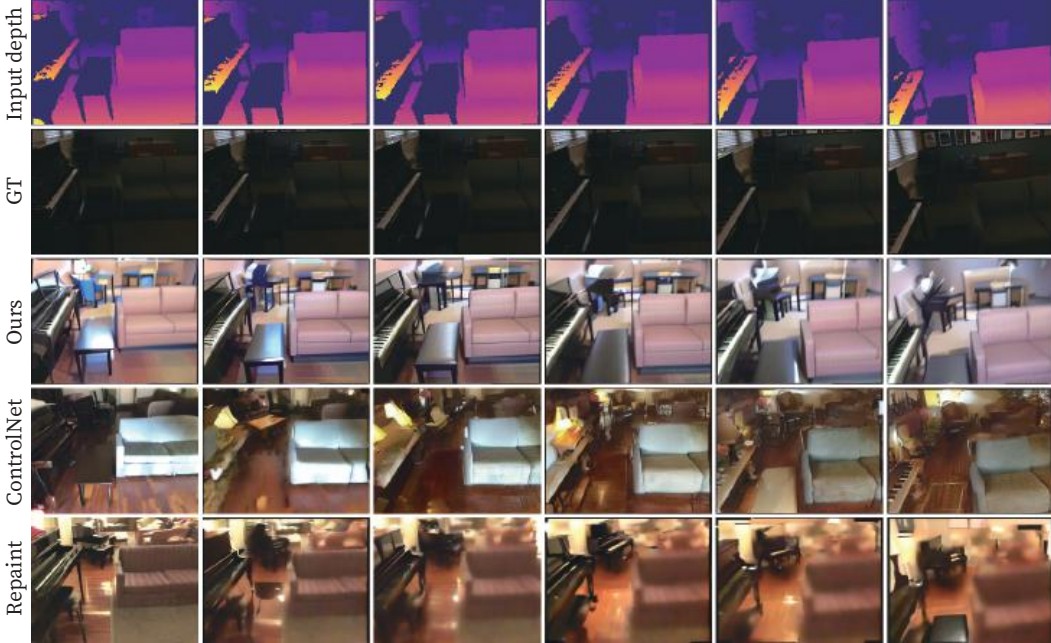

Figure 21: Addition results for depth-to-image generation.

A desk with a computer, keyboard, mouse and a teddy bear. A green stool is next to the desk.

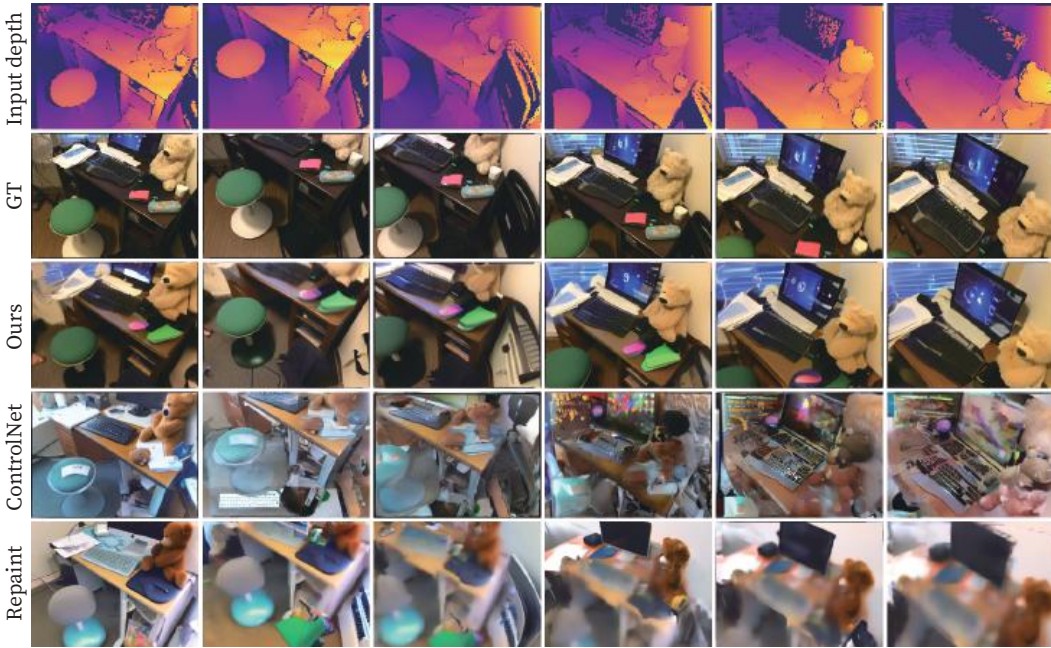

A desk with a computer and a computer monitor and a keyboard on it. A computer desk with a magazine on top of it.

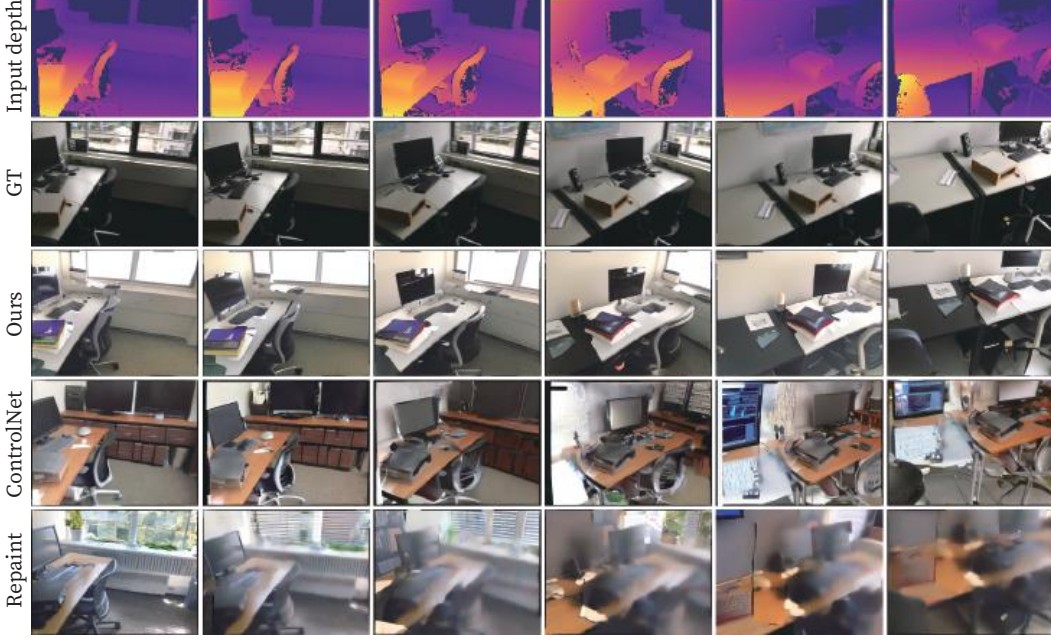

Figure 22: Addition results for depth-to-image generation.

A trash can sitting next to a radiator in a bathroom. A bathroom with a toilet and a roll of toilet paper.

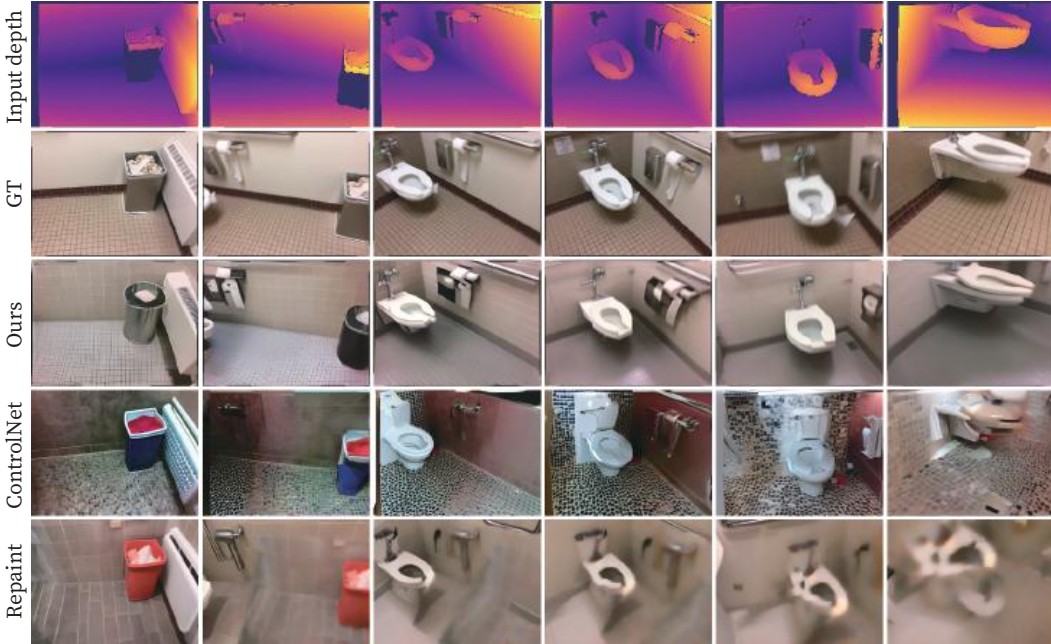

A bedroom with a bunk bed and a desk. A bunk bed with a desk underneath it.

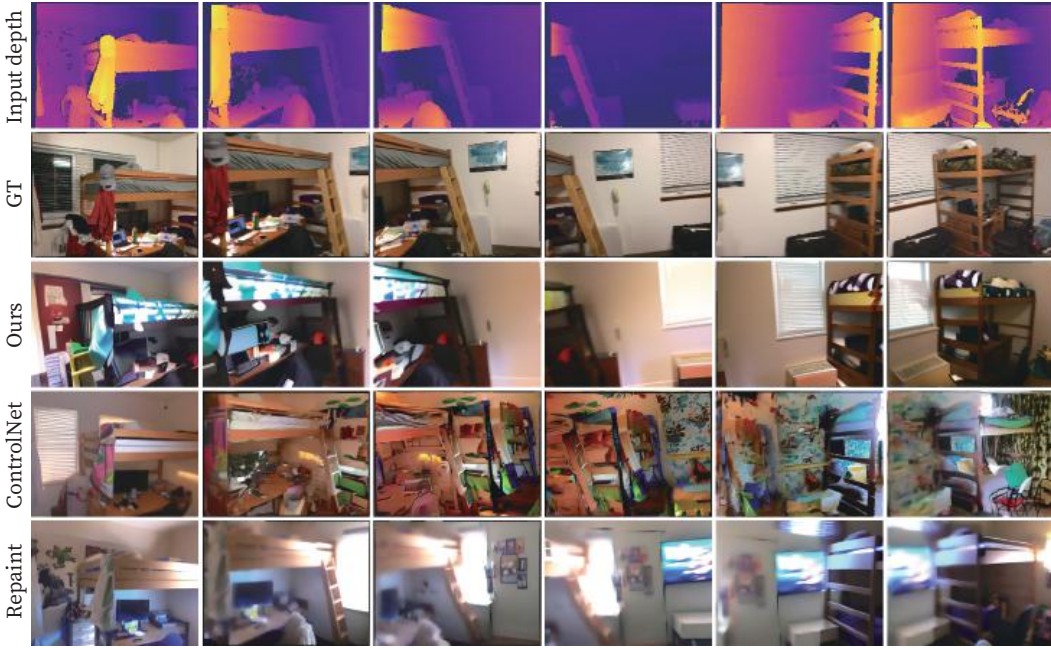

Figure 23: Addition results for depth-to-image generation.

A flat screen TV sitting on top of a tv stand. A room with a standing speaker and a TV.

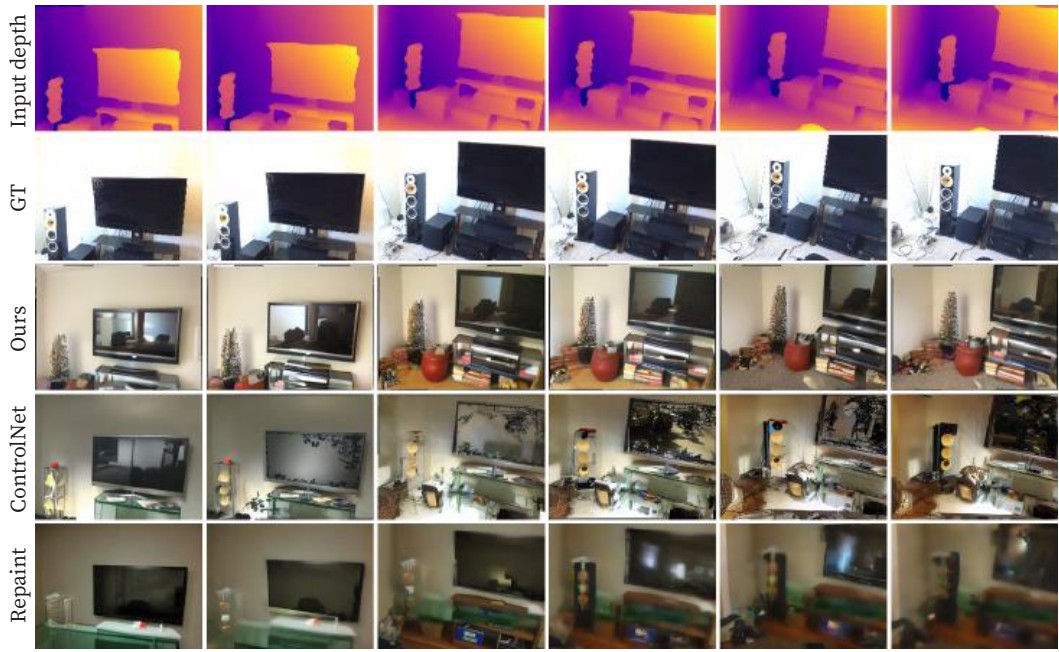

A laptop computer sitting on top of a desk next to a bed. a pair of shoes sitting on the floor next to a bed.

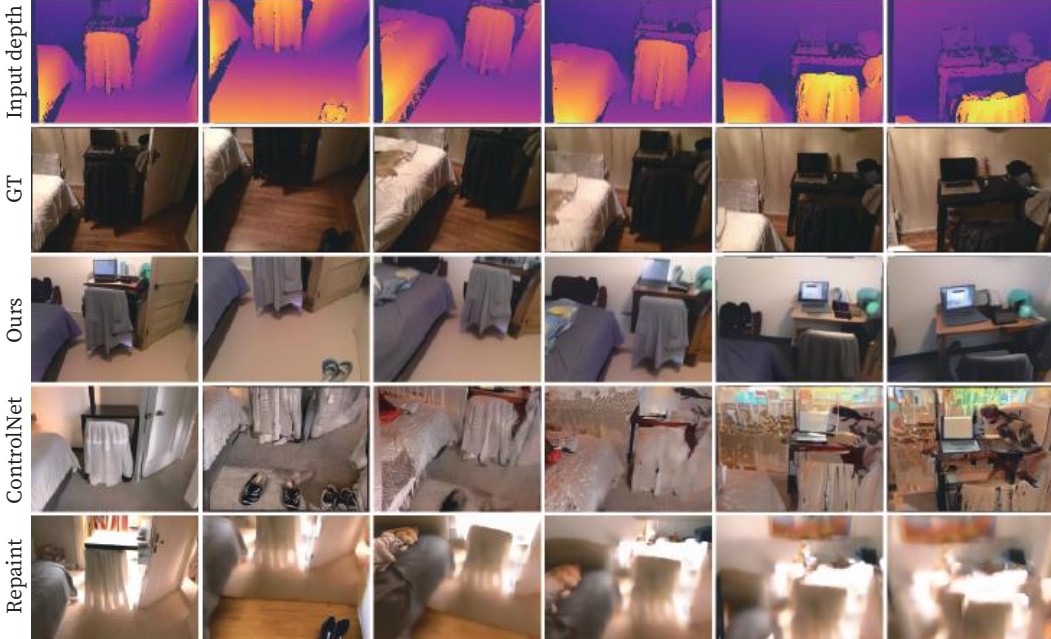

Figure 24: Addition results for depth-to-image generation.

A kitchen with a sink and a refrigerator. A kitchen with wooden cabinets and a stainless steel sink.

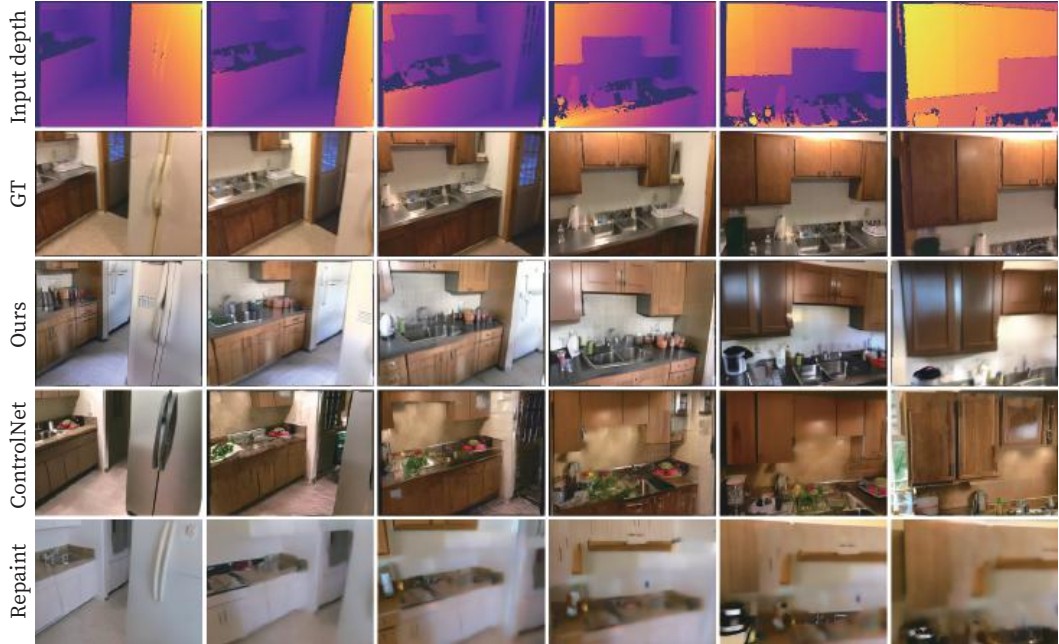

A brown couch sitting in a living room next to a window. A bookshelf filled with lots of books next to a window.

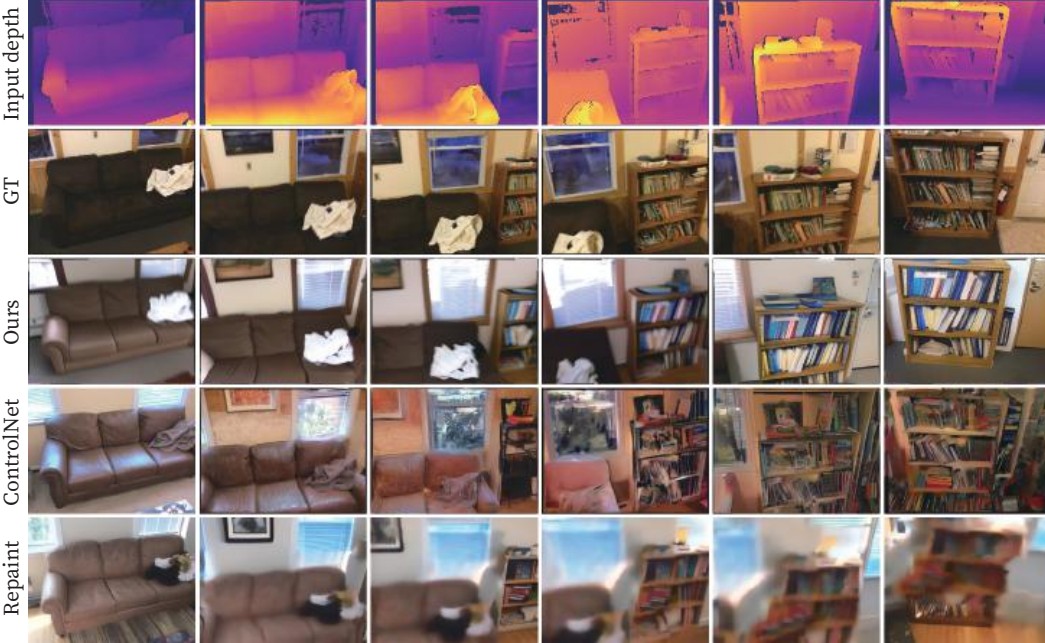

Figure 25: Addition results for depth-to-image generation.

A bedroom with a bed, desk and chair. A desk with a computer and a chair in a room. A desk with a chair and a lamp on it.

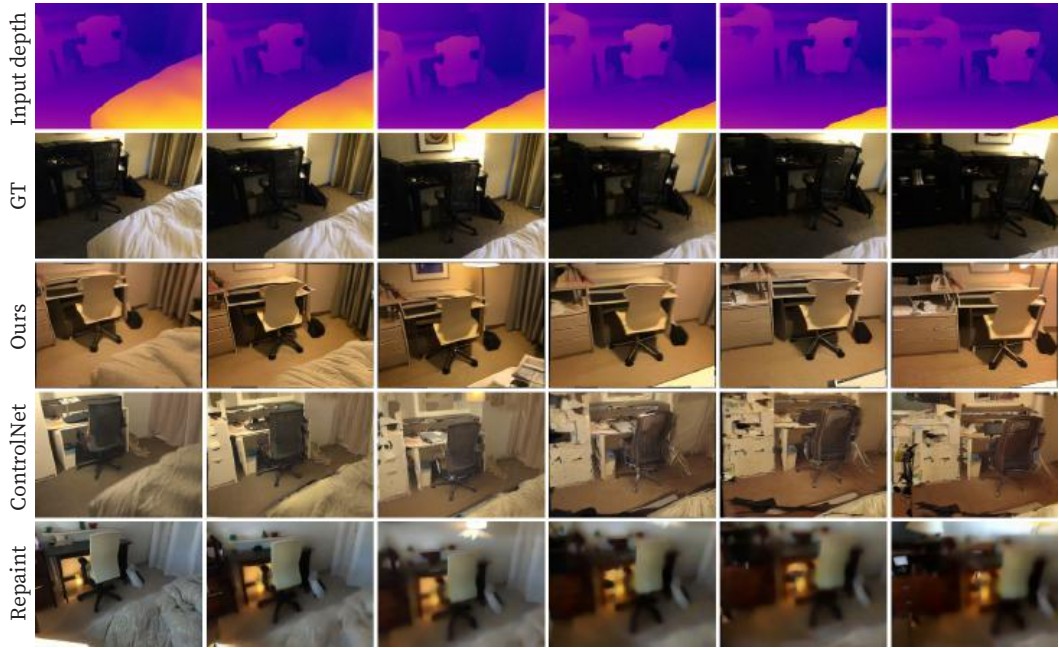

A desk with a chair and a computer monitor in a room. Two trash cans are on the floor.

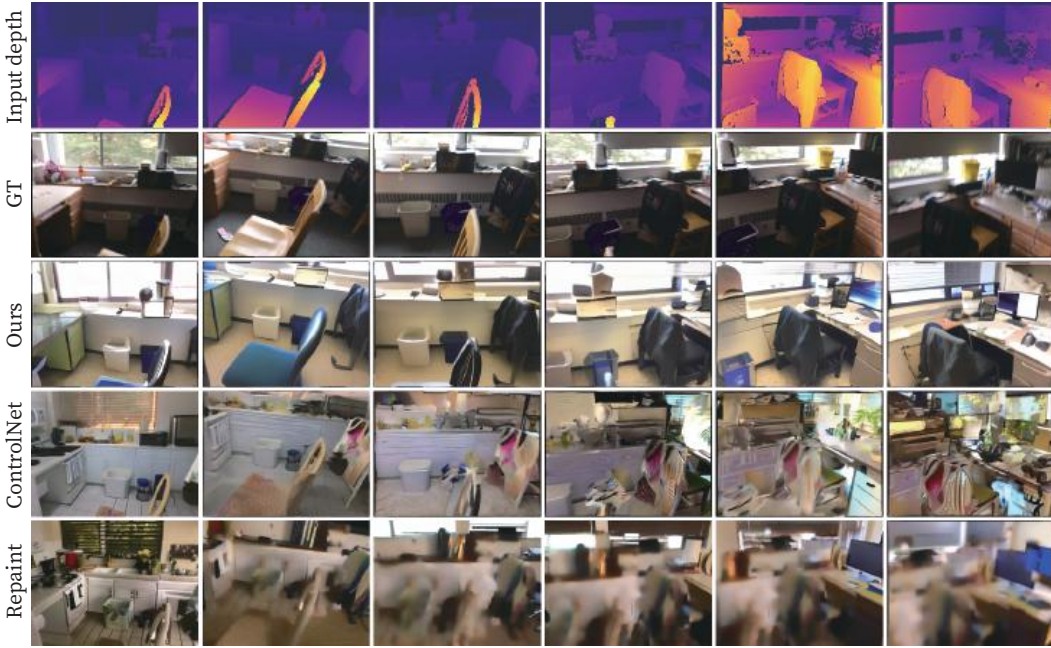

Figure 26: Addition results for depth-to-image generation.

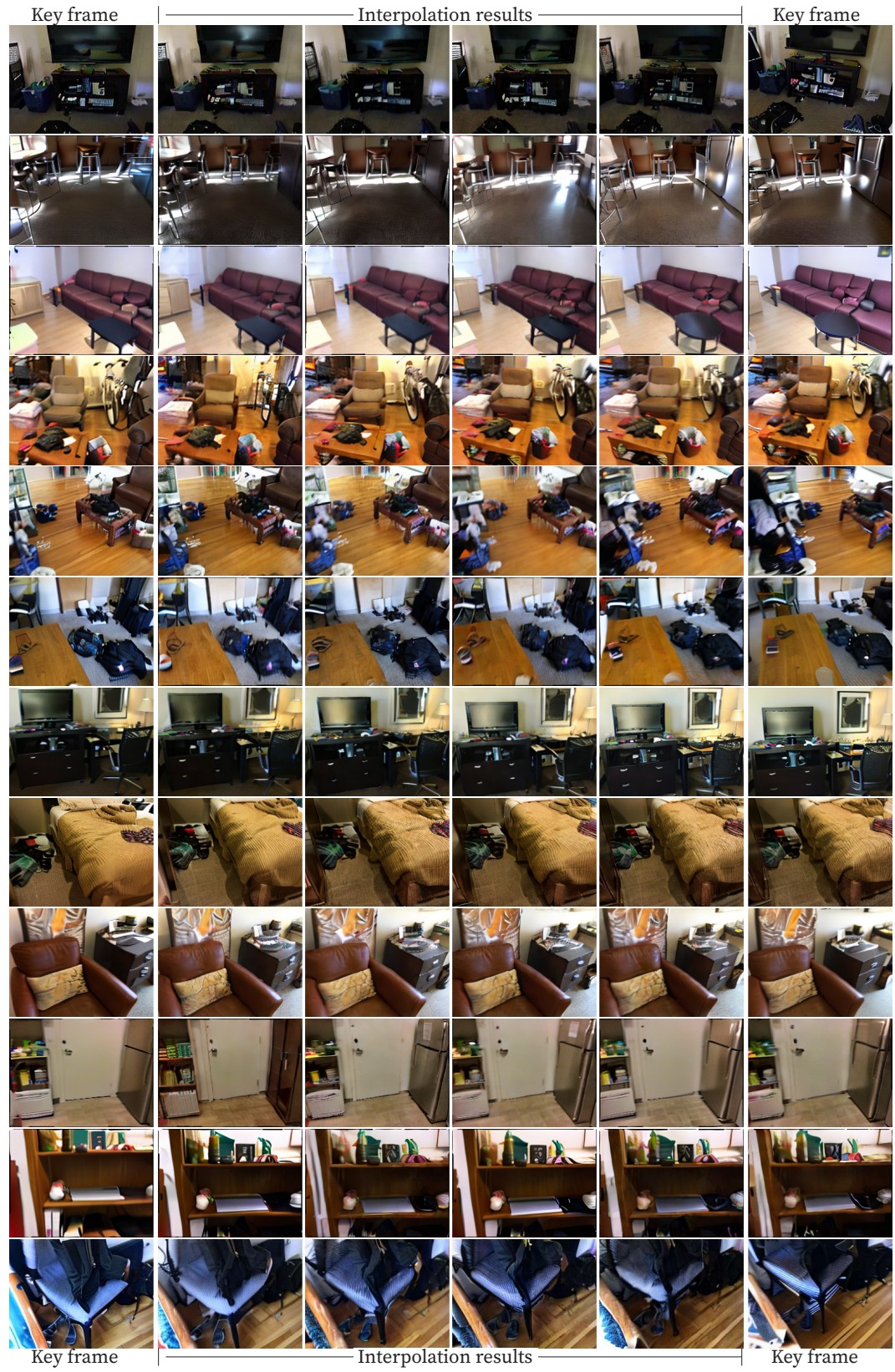

Figure 27: Addition results for interpolated frames.

