# OpenReview forum: "MVDiffusion: Enabling Holistic Multi-view Image Generation with Correspondence-Aware Diffusion"
_NeurIPS.cc/2023/Conference — NeurIPS 2023 spotlight_

### Official Review · Reviewer_VE1c · 2023-07-04

**Soundness:** 3 good
**Presentation:** 3 good
**Contribution:** 3 good
**Rating:** 7
**Confidence:** 4

**Summary:**

This paper proposes a new method to generate multi-view images, ensuring pixel-to-pixel multi-view consistency.
The multi-view consistency is guaranteed by the correspondence aware module, which is utilized in the latent diffusion multi-image generation process.
The proposed method outperforms previous works for tasks related with multi-view image generation, especially in the multi-view consistency.

**Strengths:**

1. The idea to guarantee pixel correspondence in the diffusion process is novel and reasonable.
2. The supplementary material provides codes, which show some details of the implementation.
3. The proposed method may improve the results for text to texure given meshes.
4. For different view images, the user can input different prompt.

**Weaknesses:**

1. To obtain the pixel correspondence for multi-view images, the proposed method requires to input the depth maps.
2. The authors do not explain how to deal with the depth occlusion and how to deal with the situation where the pixels in target images are not in the view of the source image.

**Questions:**

1. How does the proposed method perform for objects? Could we use the proposed method to generate textures given object meshes?
2. How does the authors deal with the depth occlusion and the situation where the pixels in target images are not in the view of the source image?

**Limitations:**

The authors addressed the limitations and the work does not have negative societal impact.

---

> ### Author Rebuttal · Authors · 2023-08-04
>
> We thank Reviewer VE1c for constructive suggestions.
>
> ---
> **W1:** Our current method is restricted to the generation of multi-view images where one-to-one correspondences are readily available. Unfortunately, this condition isn't fulfilled in casual multi-view image generation scenarios, and thus our method requires depth inputs. Developing methods that can accommodate casual multi-view image generation without the necessity of one-to-one correspondence should be an important direction for future work.
>
> ---
> **W2:** We have enhanced the positional encoding (Figure 3 in the main paper) used in the correspondence-aware attention by incorporating depth differences between source and target views. Specifically, we project the depths of correspondences from the target views onto the source views and subsequently calculate the depth differences. Notably, in the case of occlusions, we do not apply any specific handling. We anticipate that the standard SD modules will interpolate or inpaint in the presence of occlusions by learning from the depth differences. For areas of occlusion where two images lack overlap, there's no requirement to ensure consistent outcomes. We updated the results in the global rebuttal's PDF file. Please refer to the *GR1 of the general response* for updated results on multiview depth-to-image generation.
>
> ---
> **Q1:** We believe that MVDiffusion is able to generate textures given object meshes since our method can generate textures of a scene, as shown in the PDF file of the global response. Please refer to the *GR1 of the general response*.
>
> ---
> **Q2:** Please see the *GR1 of the global response* for our answers.

---

### Official Review · Reviewer_xhAT · 2023-07-06

**Soundness:** 4 excellent
**Presentation:** 3 good
**Contribution:** 3 good
**Rating:** 6
**Confidence:** 5

**Summary:**

This paper introduces MVSDiffusion, a diffusion framework that generates multi-view images with content consistency. The problem setting is interesting and essential in practice. The proposed correspondence-aware attention mechanism provides cues for multi-view consistency.

**Strengths:**

1. The paper proposes a novel problem setting: generating images for multiple camera poses with consistent contents.
2. The paper proposes a multi-view diffusion technique to obtain content consistency: extract features from other views guided by depth maps.
3. Panorama generation can also be improved via the multi-view diffusion framework.

**Weaknesses:**

1. The MVDiffusion relies on given high-quality multi-view depth maps. However, the depth map obtained for leaves and strings may lack details, which may impact the generated image quality.
2. The correspondences are derived from wrapped depth so that it can only support rigid static scenes.
3. The interpolation module is time-consuming but the time consumption of each module is not provided. Please show them.
4. The tailored super-resolution module is not novel.


**Questions:**

1. The presented panorama is blurry compared with SD (pano). Why?
2. From my point of view, MVDiffusion is able to generate multi-view images and thus can synthesize images in 3D novel views instead of pure rotations (panorama). But all the presented image sequences are almost pure rotation. What's the problem?


**Limitations:**

1. The multi-view image generation relies on given geometries.
2. Regions that are not captured by the pre-defined camera poses cannot be recovered.
3. As the framework needs to interpolate camera poses between keyframes. The time consumption may be quite large.

---

> ### Author Rebuttal · Authors · 2023-08-04
>
> We thank you for the constructive feedback and suggestions. We reply to the questions/concerns in the following:
>
> ---
> **W1.** While our experiments are currently with the ScanNet dataset, we believe that the most pragmatic usage is the creation of textures for handcrafted scene meshes. These meshes typically possess flawless geometries. This holds significant potential for real-world applications, particularly in design, visual effects, and entertainment industries.
>
> ---
> **W2.** Yes. Right now, our method only applies to static scenes, but non-rigid scenes should be future work since we can still obtain correspondences through non-rigid transformation.
>
> ---
> **W3.** We measure the run time on a single A6000 GPU. For panorama, we measure the run time by generating 8 perspective images and we don't use the interpolation module for panorama generation. For multiview depth-to-image generation, we measure the runtime by generating 12 images for generation and super-resolution module and the runtime by generating 4 interpolated images for interpolation modules.  The time consumption is as follows, which will be added to the paper:
> | Task           | Generation model | Interpolation model | Super-resolution model |
> |------------------|:----:|:-----:|:-----:|
> | Panorama |  67s |  / | 1120s  |
> | Multiview depth-to-image generation      |  128s |  104s |  1443s
>
> ---
> **Q1.** There is still inconsistency between perspective images. When they are stitched together, the panorama becomes blurry. The reason is that we use a resolution of 256 * 256 for the generation module, and thus the latent space is in low resolution, and there exists pixel inconsistency. This could be mitigated by training the generation module with a higher resolution (512 * 512). We show one example in Figure 1 of the rebuttal file, which is much clearer. Please refer to the *GR4 of the general response* for updated results of panorama generation.
>
> ---
> **Q2.** As we indicated in the abstract, our current method is restricted to the generation of multi-view images where one-to-one correspondences are readily available. Unfortunately, this condition isn't fulfilled in casual multi-view image generation scenarios. Therefore, developing methods that can accommodate casual multi-view image generation without the necessity of one-to-one correspondence is a valuable direction for future work.

---

### Official Review · Reviewer_x9w4 · 2023-07-06

**Soundness:** 3 good
**Presentation:** 3 good
**Contribution:** 3 good
**Rating:** 7
**Confidence:** 3

**Summary:**

This paper presents MVDiffusion, a new diffusion model to generate consistent multi-view images, e.g., panorama. The authors propose a novel correspondence-aware attention mechanism in order to enforce pixel-level correspondence and cross-view consistency. More specifically this mechanism is used in three modules: a generation module that generates consistent low-resolution multi-view images; an interpolation module that generates images in between; a super-resolution module. After injecting such module into Stable Diffusion Unet layers and fine-tuning the model on multi-view image dataset, the model can synthesize consistent multi-view images based on text or depth.

**Strengths:**

1. The main idea (improving multi-view consistency through a correspondence-aware attention mechanism) is novel, simple, effective and easy to understand. The generated results are very impressive, see figure 1, 4, 5, 6 and all the images in the supplementary.
2. The paper is well-written especially the method part where the main design is demonstrated clearly, see section 4.1 and figure 3. The authors also provide the code in the supplementary.
3. The comparison with previous methods and some straightforward baselines are comprehensive and the improvements are convincing.

**Weaknesses:**

1. The pipeline figure (Figure 2) could be further improved: this figure is supposed to make the audience understand the whole workflow without looking at the method text, however, for now, in this figure, it's a bit unclear to me how these modules are associated with each other at first glance.
2. Adding a brief section on failure cases would be great: the results in the main paper and supplementary are truly impressive, however, I'm also curious what are the common failure cases of MVDiffusion. Adding a short paragraph discussion this would be beneficial to the whole community.
3. I feel a bit confused about the panorama generation during inference time: in Line 145, it says "The generation module generates eight 256 × 256 images, and the super-resolution module upscales to eight 1024 × 1024 images". I am just wondering where's the interpolation module then? Or let me put this way, how many images are generated by the generation module and how many are generated by the interpolation module? I assume the key frames are generated by the generation module and more images in between are generated by the interpolation module?
4. How to handle the conflicts in the overlap region between two generated images? If I understand correctly, although correspondence attention works very well, it still cannot guarantee the overlap region between two generated images are *exactly* matched, so I'm just curious how does this mismatch is handled? I asked this because in the stitched 8 perspective images, it looks very consistent.
5. Three modules are proposed in this paper, it would be great if an ablation study could be conducted such that the audience could understand the influence of each individual module better, e.g., how much difference it would make if removing the interpolation module?
6. In addition to the stitch image, it would be great if showing more *densely* (much more than 8) generated multi-view images and make it an animation, such that the multi-view consistency could be better evaluated qualitatively by the audience.


**Questions:**

It would be great if the authors could respond the points mentioned above. Thanks!

---

> ### Author Rebuttal · Authors · 2023-08-04
>
> We thank Reviewer x9w4 for the valuable questions and suggestions. We address the comments in the following:
>
> ---
>
> **W1:** Thanks for your suggestion. For the final version, we plan to enhance Figure 2. We will illustrate a pipeline flow to show that images are initially created by the generation module, followed by the interpolation module (multiview depth-to-image generation), and finally by the super-resolution module. Additionally, we will include a note indicating that the interpolation module does not participate in the panorama generation.
>
> ---
> **W2:** Considering our model allows users to dictate distinct prompts for each perspective image, significant content change can emerge when these prompts undergo drastic alterations. Please refer to *GR4 of the general response* for examples of failure cases. We will add more examples in the final version of the paper.
>
> ---
> **W3:** In our panorama generation process, we do not employ the interpolation module. The reason for this is that the 8 perspective images we use already encompass a full 360-degree view without any occlusions. Therefore, the use of interpolation to fill in gaps or unseen areas becomes unnecessary. Conversely, the interpolation module is vital in multiview depth-to-image generation, when dealing with a large number of multi-view images. In cases where the quantity of images exceeds the memory capacity of a single GPU, the interpolation module steps in to effectively manage the images. We will clarify this in the final version.
>
> ---
> **W4:** In our approach, we do not explicitly address conflicts, as our objective is for the diffusion model to learn to generate perfectly consistent images. In panorama generation, we don't see obvious inconsistencies. However, inconsistencies do occur in the multiview depth-to-image generation. We believe these inconsistencies could arise due to two key factors. Firstly, the presence of noisy depth data can introduce errors into the generation process, leading to inconsistencies in the output images. Secondly, the employed latent space might not be high-resolution enough. This lack of granularity could result in the loss of pixel-level consistency, since the attention operation is solely conducted within the latent space.
>
> ---
> **W5:** If we remove the interpolation module, we cannot generate images that cover the whole scene mesh, as presented in Figure 2 of the rebuttal file. As for the ablations on the super-resolution module, we are not able to present qualitative results in the single-page rebuttal PDF due to the space constraint, but we will include it in the final version.
>
> ---
> **W6:** We have made the video, and the link has been sent to AC as guided by the emailed instructions.

---

> > ### Comment · Reviewer_x9w4 · 2023-08-17
> > **thanks for the detailed response**
> >
> > Thanks a lot to the authors for the detailed response! I don't have more questions but just curious where I am able to see the video? (I think this is supposed to be a question for the area chair instead of the authors then).

---

> > > ### Author Response · Authors · 2023-08-17
> > >
> > > I have informed AC to send the link.

---

> > > > ### Comment · Area_Chair_5kjR · 2023-08-21
> > > > **Video link**
> > > >
> > > > Here is that link: https://anonymous.4open.science/api/repo/submission_anonymous-06F6/file/rebuttal_video_prompt.mp4

---

### Official Review · Reviewer_2y4s · 2023-07-10

**Soundness:** 3 good
**Presentation:** 3 good
**Contribution:** 3 good
**Rating:** 5
**Confidence:** 3

**Summary:**

The paper proposes a multiview latent diffusion model that is aware of the correspondence between views.  Equipped with correspondence-aware attention blocks, the proposed generation module, interpolation module and the super-resolution module help MVDiffusion outperforms existing works.

**Strengths:**

The proposed Correspondence-Aware Attention block connects different views. Also, keyframes are first generated and then upsampled both spatially and temporally.

**Weaknesses:**

My biggest concern of the paper is the evaluation of measuring multi-view consistency. On L217 the authors mentioned to use pixel-level similarity for multi-view consistency. Is the PSNR computed between generated image and ground truth image? Does that mean the generation of depth-to-image is deterministic? It might be a good idea to conduct some evaluation in 3D. For example, train a NeRF with generated images and evaluate the difference between rendered image and the MVDiffusion generated image. Another way might be calculating some reprojection error since we know the camera poses of the generated images.
Also, for panorama generation, the author only compares with Stable Diffusion. Would be nice if DiffCollage or MultiDiffusion is also considered for comparison.

**Questions:**

Please refer to weakness.

**Limitations:**

Please refer to weakness.

---

> ### Author Rebuttal · Authors · 2023-08-04
>
> We thank Reviewer 2y4s for the constructive suggestions and feedback. We answer the questions as follows:
>
> ---
> **1. Is the PSNR computed between generated image and ground truth image?**
>
> No. The PSNR is calculated between two consecutive generated images in their overlapping regions, L219 in the main paper, since the generated image is different from ground truth image (not deterministic).
>
> ---
> **2. It might be a good idea to conduct some evaluation in 3D. For example, train a NeRF with generated images and evaluate the difference between rendered image and the MVDiffusion generated image.**
>
> We conducted evaluations in 3D using TSDF fusion to integrate the depths and images into a mesh. For experiment details, please refer to the *GR1 of the general response* for new qualitative results.
>
> ---
> **3. Would be nice if DiffCollage or MultiDiffusion is also considered for comparison?**
>
> Please see the *GR2 of the general response* for a qualitative comparison with MultiDiffusion or DiffCollage.

---

### Author Rebuttal · Authors · 2023-08-04

We thank all reviewers and appreciate the constructive comments and the recognition of novelty, and we are grateful for all the positive initial ratings (two accept, one weak accept, one borderline accept).

This general response provides updated figures and accompanying discussions to answer several key comments/questions from the reviewers. We also uploaded a video showing a rotating camera view, as requested by reviewer x9w4. The link to the video has been sent to AC, following the emailed instructions. We encourage every reviewer to watch this video.

We first summarize the key comments/questions covered in this response in the following:

---
### Summary of key questions/comments:

**S1.** Reviewer 2y4s asks for evaluating MVDiffusion in 3D.

**S2.** Reviewer 2y4s asks for a comparison with MultiDiffusion or DiffCollage.

**S3.** Reviewer x9w4 asks for failure cases of MViffusion.

**S4.** Reviewer xhAT asks why the presented panorama is blurry compared with SD (pano).

**S5.** Reviewer VE1c asks how MVDiffusion deals with the depth occlusion.

---

We will then address the comments/questions summarized above by referring to the new experimental results attached in the single-page PDF:

**GR1. (for S1 and S5) Updated results on multiview depth-to-image generation.**

In our rebuttal, Figure 2 showcases the ability of MVDiffusion to produce high-quality textures for scene meshes. The following modifications have been implemented: 1) We improved the positional encoding used in multiview depth-to-image generation, as depicted in Figure 3 of the main paper. This was achieved by integrating depth disparities between source and target views. To be precise, we project the depth of correspondences from the target views onto the source views and then compute the resulting depth disparities (handle occlusions), 2) During the training phase, every training sample incorporates 12 consecutive keyframes, 3) During testing, we first employ the generation module to produce all the key frames within a given test sequence. Then, the interpolation module is utilized to enrich or densify these images. Notably, even though our model has been trained using a frame length of 12, it has the capability to be generalized to accommodate any number of frames, and 4) Ultimately, we fuse the RGBD sequence into a cohesive scene mesh.

---
**GR2. (for S2) Comparison with MultiDiffusion or DiffCollage.**

As pointed out in Line 64 of the paper, neither MultiDiffusion nor DiffCollage incorporates a camera model when generating 360-degree views. This leads to outputs that do not accurately represent true panoramas. In our rebuttal's Figure 5, we present the results of MultiDiffusion and MVDiffusion using the same prompts. In a true panorama, lines often appear curved due to the distinct perspective shifts inherent in panoramic photography. However, in images produced by MultiDiffusion, these lines remain linear, retaining the characteristics of a conventional perspective image

---
**GR3. (for S4) Updated results of panorama generation**
We trained the correspondence-aware attention block within our panorama generation module at a resolution of 512 x 512 while keeping the original stable diffusion UNet weights frozen using the Matterport3D indoor datasets.  As shown in Figure 1, our method is able to generate outdoor panoramas. The result indicates: 1) Even when trained on a limited dataset, MVDiffusion exhibits strong generalization capabilities due to the stable diffusion pretrained model, and 2) The clarity of the generated panorama is notably enhanced—with fewer artifacts—when trained at higher resolutions.

---
**GR4. (for S3) Failure cases of panorama generation**

In Figure 3, we highlight a failure case of our panorama generation. Considering our model allows users to dictate distinct prompts for each perspective image, significant content change can emerge when these prompts undergo drastic alterations.

---
**GR5. At last, we update a new functionality of MVDiffusion, which supports extrapolating a perspective image to a full 360 view.**

We also demonstrate our method can extrapolate a perspective image into 360 views in Figure 4.  Specifically, we use the pretrained stable diffusion impainting model as the base generation model. In the Unet branch of the conditioned image, we append a mask of ones (in 4 channels) to the image. This concatenated image then serves as the input for the inpainting model, which ensures the content remains consistent. Conversely, in the Unet branch dealing with the generated images, we concatenate a black image (pixel values of zero) with a mask of zeros. This serves as the input, enabling the inpainting model to generate a completely new image based on the text it is provided.

---

### Decision · Program_Chairs · 2023-09-21

**Decision:**

Accept (spotlight)

**Comment:**

This paper appears was unanimous accept by the reviewers. They all appreciated the novelty and thorough evaluation and felt the paper is a good contribution to the field.  The author rebuttal was reviewed and commented on by all with healthy discussion between the authors and the reviewers and the AC.  Congratulations!

In revising your paper for the camera-ready, please include and comments and clarifications that were made or proposed during the rebuttal and discussion phase.